# Zero-source LLM Hallucination Detection with Human-like Criteria Probing

**Jiahao Yang** [* 1 2]  **Shuhai Zhang** [* 1 3]  **Hailong Kang** [* 1]  **Feng Liu** [2]  **Qi Chen** [4]  **Mingkui Tan** [1 3]

## Abstract

Large language models (LLMs) often hallucinate by generating factually incorrect or unfaithful content, posing significant risks to their safe use. Detecting such hallucinations is particularly challenging under the *zero-source constraint*, where no model internals or external references are available, and detection must rely solely on the textual query–answer pair. In this paper, we propose *Human-like Criteria Probing* for Hallucination Detection (HCPD), a paradigm that emulates the multi-faceted reasoning of human evaluators. Its core is a *Human-like Criteria Probing* (HCP) mechanism, in which a LLM agent adaptively decomposes its judgment into a weighted set of interpretable criteria and aggregates criterion-specific scores into a final truthfulness measure. To achieve this adaptive capability, we introduce a reward-based alignment scheme using only weak supervision from semantic consistency. At inference, we employ a multi-sampling aggregation strategy to ensure robust decisions while preserving full interpretability. We further provide theoretical analysis supporting the reliability of our approach. Extensive experiments show that HCPD consistently outperforms state-of-the-art baselines, offering an effective and explainable solution for zero-source hallucination detection. Code is available at `https://github.com/TRISKEL10N/HCPD`.

## 1. Introduction

Large language models (LLMs) have rapidly advanced and are increasingly deployed across a broad range of applications, including information retrieval (Zhu et al., 2025),

---
[*]Equal contribution [1]South China University of Technology [2]The University of Melbourne [3]Pazhou Laboratory [4]Australian Institute for Machine Learning, Adelaide University. Correspondence to: Mingkui Tan <mingkuitan@scut.edu.cn>, Qi Chen <qi.chen04@adelaide.edu.au>, Feng Liu <fengliu.ml@gmail.com>.

*Proceedings of the 43rd International Conference on Machine Learning*, Seoul, South Korea. PMLR 306, 2026. Copyright 2026 by the author(s).

decision support (Chiang et al., 2024; Chen et al., 2024b; Ma et al., 2025), and domain-specific assistance in healthcare (Benary et al., 2023; Vrdoljak et al., 2025), finance (Yu et al., 2024), and education (Neumann et al., 2024). Nevertheless, their practical use is constrained by hallucinations, where LLMs generate responses that are factually incorrect, ungrounded, or unfaithful to the user's intent, posing significant risks in safety-critical settings. Consequently, reliable hallucination detection has become essential for the safe and trustworthy deployment of LLM-based assistants.

A critical challenge is that practical hallucination detection frequently operates under a strict **zero-source** constraint (Fang et al., 2025; Yang et al., 2025b). In prevalent open-world scenarios, the auditing process is entirely decoupled from the generation. For instance, third-party auditors (e.g., social media platforms, news agencies) must evaluate massive user-uploaded texts without knowing the underlying source LLMs. Similar situations also occur when the vast majority of end-users interact with LLMs through web interfaces (*e.g.*, ChatGPT[1], Gemini[2] and Claude[3]) or browser extensions, where plain text is the sole accessible output. Consequently, commercial APIs, internal states, and auxiliary resources (*e.g.*, external knowledge bases) are typically unavailable. Under such realistic restrictions, robust detection must rely solely on the observed query–answer pair.

Unfortunately, most existing approaches are not directly applicable under the aforementioned constraint. While effective in the traditional knowledge-based hallucination detection, retrieval-augmented or fact-verification methods (Semnani et al., 2023; Hu et al., 2024; Chen et al., 2024c) require access to web or knowledge resources, whose availability and reliability are difficult to guarantee. Avoiding external references, confidence-based and metric-based methods (Malinin & Gales, 2021; Kuhn et al., 2023; Park et al., 2025) mainly depend on model internals, which are unattainable for black-box or commercial systems. Self-supervised or consistency-based methods (Kadavath et al., 2022; Manakul et al., 2023) typically employ static, task-agnostic heuristics, limiting their ability to capture the precise, context-dependent judgments across diverse domains. Moreover,

---
[1]`https://chatgpt.com`
[2]`https://gemini.google.com/app`
[3]`https://claude.ai/new`

most detectors provide only binary labels or scalar scores, offering limited interpretability and diagnostic insight.

In contrast, human experts rarely judge a response using a single monolithic criterion. They instead decompose evaluation into multiple dimensions, adapt their relative weights to the context, and provide evidence-grounded judgment. This observation motivates a general *zero-source detection paradigm* that emulates human-style evaluative reasoning.

In this paper, we propose **Human-like Criteria Probing for Zero-source Hallucination Detection (HCPD)**. Its core is an *Human-like Criteria Probing (HCP)* mechanism, which enables a pre-trained LLM agent to evaluate responses through a transparent, multi-step process (Section 4.3). For each query–answer pair, the agent first adaptively generates a set of fine-grained criteria (*e.g.*, factual accuracy, logical consistency) and their context-aware importance weights, then scores the text against each criterion, and finally aggregates these scores into an overall truthfulness measure, effectively emulating the nuanced, multi-perspective reasoning of a human expert. To enable this adaptive judgment capability, we introduce a *reward-based alignment training* scheme, which leverages weak supervision from semantic consistency to teach the agent how to decompose and weight criteria without needing ground-truth hallucination labels (Section 4.4). At inference, we apply a *multi-sampling aggregation* strategy to reduce variance from the stochastic generation process, performing $K$ independent HCP evaluations per instance and averaging the results for a robust final decision (Section 4.5). We further provide a statistical characterization of both training and inference behaviors, and derive a threshold-free performance characterization by bounding the ranking error probability (Section 4.6). Extensive experiments demonstrate that HCPD outperforms existing state-of-the-art methods, validating the effectiveness under the zero-source hallucination detection.

Our contributions are summarized as follows:

- An adaptive, multi-criteria probing framework for zero-source detection: We are the first to explicitly formalize hallucination detection under zero-source constraint, and propose Human-like Criteria Probing mechanism, which reframes detection as a process of context-aware criteria generation, weighting, and aggregation. By enabling an agent to adaptively decompose its judgment into interpretable dimensions, our method emulates the nuanced, multi-faceted reasoning of human evaluators, moving beyond monolithic scoring paradigms.

- Weakly-supervised alignment training without ground-truth labels: To achieve reliable adaptive judgment in the scoring agent, we introduce a reward-based alignment training scheme using only weak supervision derived from semantic consistency. This method effectively teaches the agent to identify, weight, and

score relevant criteria without requiring any annotated hallucination data, making it uniquely suited for the practical zero-source constraint.

- A stable and interpretable inference strategy with theoretical guarantees: We design a multi-sampling aggregation strategy during inference to mitigate generation variance, enhancing decision robustness while preserving full interpretability through the generated criteria and weights. Furthermore, we develop a theoretical analysis, providing formal insights into the feasibility and reliability of our probing-based approach.

## 2. Preliminaries

**Group Relative Policy Optimization.** Group Relative Policy Optimization (GRPO) is a reinforcement learning algorithm designed for the stable and efficient fine-tuning of LLMs. To obviate the need for an explicit value function commonly required in policy gradient methods (Schulman et al., 2017), GRPO adopts the average reward of multiple sampled outputs conditioned on the same prompt as an implicit baseline. Concretely, for each input $x$, the current policy $f_\theta$ samples a group of outputs $\{Y_1, \ldots, Y_G\}$ and assigns each output a critic-based scalar reward $r(Y_g)$. Rather than optimizing the absolute reward, GRPO constructs a group-relative advantage $A_g = r(Y_g) - \frac{1}{G}\sum_{j=1}^{G} r(Y_j)$ within the group, which normalizes rewards within the group and inherently reduces variance of the gradient estimates. The policy model is then optimized to increase the likelihood of outputs with above-average rewards, while constraining updates to stay close to a reference policy $f_0$ (typically the initialization) to preserve generation quality. The objective for a single group is defined as:

$$\mathcal{J}(\theta) = \frac{1}{G}\sum\nolimits_{g=1}^{G}\left[\frac{f_\theta(Y_g|x)}{f_0(Y_g|x)}A_g\right] - \beta \cdot D_{KL}(f_\theta(\cdot|x)||f_0(\cdot|x)),$$
(1)

where $\beta$ is a hyperparameter controlling the strength of the Kullback–Leibler divergence (Csiszár, 1975) penalty. For simplicity, we reuse $\mathcal{J}(\theta)$ to denote the objective aggregated over the distribution when used in theoretical bounds.

By evaluating and comparing multiple outputs for the same prompt, GRPO obtains a robust, context-aware learning signal and achieves more stable convergence compared to per-sample absolute reward optimization. This property makes it particularly suitable for aligning our scoring agent, where rewards are dense yet require precise calibration across diverse evaluation properties.

## 3. Related Work

**Hallucination Detection.** Although truthfulness is a fundamental requirement of language generation, large language

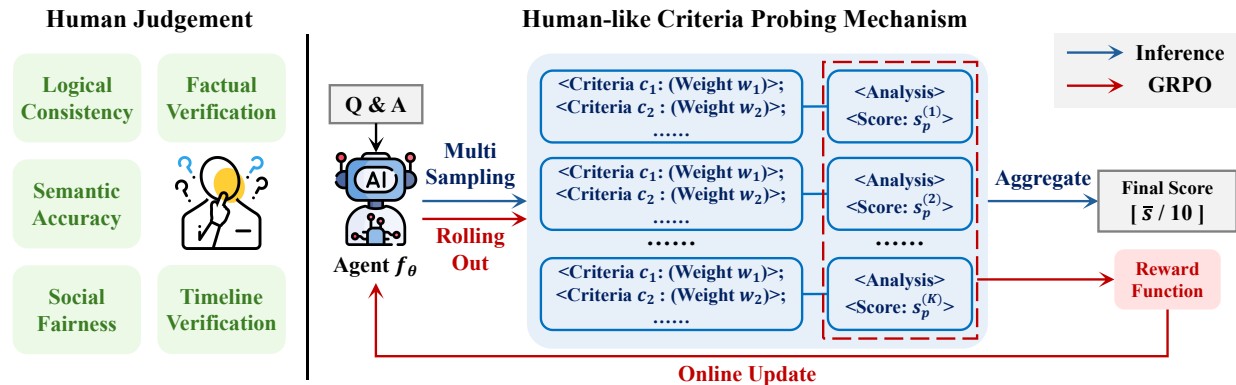

Figure 1. Overview of the proposed HCPD. Given a query–answer pair $(q, a)$, the agent instantiates a set of specific criteria $\{c_i\}_{i=1}^{m}$ and corresponding importance weights $\{w_i\}_{i=1}^{m}$. The criterion-level partial scores $\{s_i\}_{i=1}^{m}$ are subsequently produced and aggregated into an overall truthfulness measure $s_p$. During GRPO training, we fine-tune the agent by maximizing the score-alignment reward that encourages the predicted $s_p$ to match weak supervision labels derived from consistency-based similarity measures. At inference time, we invoke the agent $K$ times and aggregate the resulting scores to obtain a reliable decision $\bar{s}$.

models (LLMs) still frequently produce outputs that are factually incorrect or contextually inconsistent, commonly referred to as hallucination. Consequently, hallucination detection (Lin et al., 2022; Azaria & Mitchell, 2023; Kuhn et al., 2023; Ren et al., 2023; Manakul et al., 2023; Zhang et al., 2023; Lin et al., 2024; Chen et al., 2024a; Du et al., 2024; Park et al., 2025) has emerged as a central research focus for the safe and reliable deployments.

A prevailing research paradigm attributes LLM hallucinations to predictive uncertainty. Probability-based methods quantify such uncertainty via metrics including Perplexity (Ren et al., 2023), Length-Normalized Entropy (Malinin & Gales, 2021), and Semantic Entropy (Kuhn et al., 2023). In contrast, consistency-based methods evaluate agreement across multiple sampled responses either through similarity metrics such as BERTScore (Zhang et al., 2019), ROUGE (Lin et al., 2024), natural language inference, and prompt-based self-consistency verification (Manakul et al., 2023), or via spectral analysis of response covariance matrices such as eigenvalue decomposition (Chen et al., 2024a). Verbalized-based methods elicit confidence signals by prompting models to express uncertainty explicitly in natural language, through verbalized uncertainty (Lin et al., 2022) or self-evaluation (Kadavath et al., 2022) mechanisms. Nevertheless, these uncertainty-driven signals remain limited as hallucinations frequently occur even in high-confidence generations. Whereas multi-sample evaluation partially mitigates this issue, it incurs substantial computational overhead.

A complementary perspective infers textual truthfulness by exploiting internal model states. For instance, CCS (Burns et al., 2022) extracts latent knowledge from activation patterns, SAPLMA (Azaria & Mitchell, 2023) trains classifiers directly on hidden representations, HaloScope (Du et al., 2024) identifies hallucination-related subspaces via singular value decomposition, and TSV (Park et al., 2025) introduces learnable steering vectors to adapt latent features for

improved separability. Despite their effectiveness, *these methods necessitate access to the internal model representations*, thereby limiting their applicability in real-world deployment scenarios where the underlying model architecture and training data provenance remain undisclosed.

## 4. Method

### 4.1. Problem Definition

**LLM Hallucination.** In the context of text generation, hallucination refers to contents that are either factually inconsistent with established knowledge or contextually unfaithful to the given source or query intent (Huang et al., 2025). Such errors typically arise from the model's inherent uncertainty, limitations in its learned knowledge, or failures in reasoning. Characterizing and probing these failure modes is therefore essential for the reliable deployment of LLMs.

**Hallucination Detection under Zero-source Constraint.** The goal of hallucination detection is to assess whether a model-generated response exhibits factual inconsistencies or contextual unfaithfulness (Ji et al., 2023; Farquhar et al., 2024). Formally, let $\mathcal{Q}$ denote the space of user queries and $\mathcal{A}$ the space of model outputs. For a query $q \in \mathcal{Q}$ and a generated answer $a \in \mathcal{A}$, the detection model $f$ produces a binary label $y \in \{0, 1\}$ indicating whether the response constitutes hallucinations. In this work, we focus on the **zero-source constraint**: the detector $f$ *cannot* access the LLM's internal states (*e.g.*, token probabilities) or output distribution during inference. Furthermore, no external knowledge bases or reference texts are available. It must operate solely on the textual pair $(q, a)$, making the task challenging yet aligned with practical black-box deployment.

**Challenges in Hallucination Detection under Zero-source Constraint.** Developing an effective detector under this constraint faces several challenges. *First*, without access

*Table 1.* The structured and interpretable output from the agent.

| Component | Example |
|---|---|
| Q & A | **Qes**: In which country were the 1948 Winter Olympics held? 
 **Ans**: Norway. |
| Specific Criteria | 1. **Factual Grounding** (Weight: 60%) 
 2. **Temporal Consistency** (Weight: 20%) 
 3. **Semantic Precision** (Weight: 20%) |
| Analysis | - **Factual Grounding**: The 1948 Winter Olympics were held in St. Moritz, Switzerland, not Norway. This is a clear factual error. 
 - **Temporal Consistency**: The response does not specify a year, so there are no temporal issues directly in the text. 
 - **Semantic Precision**: The response is vague and does not provide the correct information, leading to semantic distortion as the user is not given accurate details. |
| Score (1~10) | 1 |

to model-internal confidence signals (*e.g.*, token logits), the detector must rely entirely on the semantic content, requiring a deep understanding to identify subtle inconsistencies. *Second*, the absence of reference sources or external knowledge eliminates straightforward fact-checking, forcing the detector to reason intrinsically about plausibility. *Third*, hallucinations are heterogeneous; they may manifest as factual errors, logical fallacies or semantic misalignments, demanding a flexible, multi-perspective evaluation rather than a single static criterion. *Finally*, for trust and usability, decisions must be interpretable, clarifying *why* a text is deemed hallucinatory. These motivate the need for an adaptive, interpretable, and purely text-based probing mechanism.

### 4.2. Motivations and Method Overview

**Motivations.** When assessing the correctness of a text, Human evaluators naturally engage in holistic and multifaceted reasoning. They do not apply a single, rigid rule but dynamically consider various dimensions, *e.g.*, factual accuracy, logical soundness, temporal consistency, and contextual faithfulness, and intuitively weigh their importance based on the specific content. This context-dependent, multi-criteria weighting yields two practical advantages: i) *adaptivity* — the ability to focus on the most diagnostic checks for each instance, improving detection precision; and ii) *interpretability* — the ability to explain negative judgments by pointing to specific violated criteria. These observations motivate us to design a detection paradigm that emulates the human evaluative process: decomposing the holistic judgment into a set of interpretable criteria, dynamically determining their relevance (weights) based on the input, and synthesizing a final decision through a transparent aggregation.

**Method Overview.** Motivated by the above analysis, we propose *Human-like Criteria Probing for Zero-source Hallucination Detection* (HCPD), which employs a Human-like

Criteria Probing (HCP) mechanism as its core (Figure 1). Specifically, given an input pair $(q, a)$, HCP *adaptively* uses an LLM agent to generate a set of interpretable evaluation criteria along with their relative importance weights and output an aggregate truthfulness score (Section 4.3). To achieve this capability into the agent, we devise a *Reward-Based Alignment Training* scheme using weak supervision from semantic consistency, teaching the agent to decompose and weight criteria without ground-truth labels (Section 4.4). To reduce the generation randomness, we introduce a *Multi-Sampling Aggregation* strategy by performing $K$ independent probes per instance and averaging the resulting scores as the final detection statistic; this multi-sampling aggregation stabilizes the decision while preserving interpretability (Section 4.5). We also provide theoretical analysis justifying the feasibility of our probing approach (Section 4.6).

### 4.3. Human-like Criteria Probing Mechanism

Inspired by the multi-faceted nature of human evaluation, we develop **Human-like Criteria Probing (HCP)**, a structured reasoning framework that enables a pre-trained LLM to function as an *adaptive* scoring agent for hallucination detection, moving beyond conventional monolithic scoring.

**Human-like Criteria Probing.** Given the query-answer pair $(q, a)$ under evaluation, the agent $f_\theta$ produces interpretable, criterion-wise assessments. Specifically, it first adaptively generates a set of fine-grained specific criteria $\{c_i\}_{i=1}^m$ derived from predefined *General Evaluation Criteria* set $\mathcal{C}$ in broader dimensions such as factual and logical coherence. The agent then autonomously assigns context-sensitive importance weights $\{w_i\}_{i=1}^m$ to these specific criteria, emphasizing, for instance, temporal accuracy for historical queries and logical soundness for scientific explanations. Finally, it predicts a partial score $s_i$ for each weighted criterion and aggregates them into an overall truthfulness measure $s_p$, thereby mimicking the contextual and multi-faceted nature of human judgment. The operation of the agent can be formalized as:

$$s_p = \sum_{i=1}^m w_i \cdot s_i, \quad \{(c_i, w_i, s_i)\}_{i=1}^m \leftarrow f_\theta(q, a; \mathcal{C}), \quad (2)$$

where $m$ is the number of specific criteria, the weight $w_i \geq 0$ and $\sum_i w_i = 1$, the scores are integers in $\{1, \ldots, 10\}$, and the General Evaluation Criteria is given by:

$$\mathcal{C} = \{\text{Factual}, \text{Logical}, \text{Semantic}, \text{Temporal}, \text{Social}\}. \quad (3)$$

**Structured and Interpretable Output.** To ensure transparency and facilitate downstream processing, the agent is constrained to generate outputs in a strictly structured format (Table 1). For each specific criterion, the output explicitly reports its assigned weight, the corresponding evidence that supports or contradicts the response, and the associated partial score. These components are subsequently aggregated

into an overall score via weighted synthesis. This design renders full interpretability of the detector's decisions and enables fine-grained attribution of hallucinated content.

### 4.4. Reward-based Alignment Training

To equip the agent $f_\theta$ with precise evaluative capabilities, we train it with a reward-based alignment paradigm using Group Relative Policy Optimization (GRPO) (Shao et al., 2024). This aligns the agent's adaptive probing and scoring behavior with a weak supervision signal derived from semantic consistency, eliminating the reliance on expensive human-annotated hallucination labels.

**Training Data Construction and Labeling.** Our training data is constructed from standard question-answer (QA) benchmarks that pair queries with human-verified reference answers. Specifically, for each query $q$ sourced from established datasets (*e.g.*, TriviaQA (Joshi et al., 2017)), a corresponding ground-truth answer $\hat{a}$ is available. We further leverage an auxiliary LLM (*e.g.*, the test target model) to generate a set of candidate responses $\{a^{(n)}\}_{n=1}^N$ for the same query, intentionally sampling outputs that span a spectrum from factually correct to clearly hallucinated.

Following common practice in hallucination detection benchmarks, we adopt widely used consistency metrics (*e.g.*, BLEURT (Sellam et al., 2020)) in natural language processing (NLP) as a source of weak supervision labeling. Concretely, for each generated answer $a^{(n)}$, we compute its semantic consistency to the reference $\hat{a}$, yielding $\text{sim}(\hat{a}, a^{(n)}) \in [0, 1]$. Consistent with prior work (Du et al., 2024), responses with a consistency score above 0.5 are treated as relatively faithful, whereas those below are labeled as hallucinated. To obtain a graded supervision signal, we translate this continuous score to a discrete 1–10 scale:

$$s_l^{(n)} = \text{clip}(\lfloor 10 \cdot \text{sim}(\hat{a}, a^{(n)}) \rceil, 1, 10), \qquad (4)$$

where the reference answer $\hat{a}$ is assigned the maximum score of 10. This procedure yields a weakly supervised dataset $\mathcal{D} = \{(q, \{(\hat{a}, 10)\} \cup \{(a^{(n)}, s_l^{(n)})\}_{n=1}^N)\}$ that captures gradual degrees of hallucination severity and enables the agent to learn fine-grained distinctions.

**Score-alignment Reward Function.** To encourage the agent towards producing scores consistent with the weak supervision labels, we introduce a score-alignment reward function. Given an input pair $(q, a)$, the agent performs human-like criteria probing and outputs a structured evaluation containing a predicted score $s_p \in \{1, \ldots, 10\}$. The reward function assigns a scalar reward $r \in [0, 1]$ by directly comparing the predicted score $s_p$ with the label $s_l$:

$$r = \begin{cases} 1 - \dfrac{|s_p - s_l|}{9}, & \text{if the output is well-formed,} \\ 0, & \text{otherwise,} \end{cases} \qquad (5)$$

and optimize according to Eqn. (1).

The reward attains its maximum value $r = 1$ when the prediction perfectly matches the label ($s_p = s_l$) and decreases linearly with the absolute deviation between them. Notably, the score-alignment reward implicitly enforces structural constraints on the agent's output: any deviation from the prescribed format that prevents reliable extraction of a valid score results in $r = 0$; and the inherent KL (Csiszár, 1975) regularization of GRPO in Eqn. (1) limits deviation from the initial policy (Shao et al., 2024).

**Differentiable Scoring v.s. Binary Classification.** The adoption of a differentiable scoring scheme rather than binary "True/False" classification is a central design choice. **1) The graded formulation better reflects the inherently varying degrees of hallucination severity**, which ranges from minor factual imprecision to fully fabricated content. By modeling hallucination on a fine-grained scale, the agent can capture subtle deviations that binary labels necessarily collapse. **2) The reward defined in Eq. (5) induces an informative policy optimization signal**. By penalizing prediction errors proportionally to their magnitude, it yields a denser reward landscape than binary rewards, which only indicate correctness without guiding the magnitude or direction of improvement. **3) Scalar scoring enables flexible decision-making at inference time**. Unlike binary classifiers with fixed operating points, our agent's scores can be thresholded at varying levels to accommodate different precision–recall trade-offs without retraining. Collectively, this scoring-alignment reward formulation not only facilitates effective learning under weak supervision but also endows the detector with calibrated, adaptable judgment capabilities essential for effective hallucination detection.

### 4.5. Robust Inference via Multi-sampling Aggregation

Due to the inherent stochasticity of language model generation, a single evaluation from the agent may exhibit non-negligible variance. To reduce this randomness and improve decision reliability, we employ a multi-sampling aggregation strategy during inference.

Formally, the trained agent $f_\theta$ is independently invoked $K$ times for the same input $(q, a)$ pair, yielding a set of structured evaluations together with corresponding synthesized final scores $\{s_p^{(k)}\}_{k=1}^K$. We obtain a robust estimate of response truthfulness by aggregating these scores via arithmetic averaging:

$$\bar{s} = \frac{1}{K} \sum_{k=1}^K s_p^{(k)}. \qquad (6)$$

This aggregation suppresses stochastic fluctuations in individual evaluations, resulting in more reliable and stable detection outcomes. Experimental results indicate that

our method achieves substantially stronger detection performance than SelfCKGPT (Manakul et al., 2023) while incurring a similar time cost.

## 4.6. Theoretical Analysis

To establish a theoretical foundation for HCPD, we provide guarantees for both its training and inference stages. Our analysis yields three core results: (i) Theorem 1 ensures that GRPO training anchors the learned scoring behavior to the weak supervision signal in expectation; (ii) Proposition 1 justifies multi-sampling aggregation as a variance-reduction mechanism via a concentration bound; and (iii) Corollary 1 derives a threshold-free performance characterization by bounding the ranking error probability.

We first provide theoretical guarantees for the rationality and effectiveness of the training and inference framework.

**Theorem 1.** *(**Expectation alignment under training**) Let $x$ denote an input and $s_l(x)$ its corresponding weak label. Consider the $\ell_1$-risk of predicting score $s$: $R_x(s) \triangleq |s - s_l(x)|$, whose minimizer is $s = s_l(x)$. Let $Y \sim f_\theta(\cdot \mid x)$ be a stochastic well-formed generation of the agent, and let $S_\theta(x) = s_p(x, Y)$ denote the parsed score. Define the conditional mean score as $\mu_\theta(x) \triangleq \mathbb{E}\big[S_\theta(x) \mid x\big]$, we have*

$$\mathbb{E}_x\big[|\mu_\theta(x) - s_l(x)|\big] \leq \mathcal{J}'(\theta), \qquad (7)$$

*where $\mathcal{J}'(\theta)$ is affine-equivalent to the GRPO objective $\mathcal{J}(\theta)$ defined in Eqn. (1).*

Theorem 1 formalizes a training-time alignment guarantee: optimizing the KL-regularized GRPO objective forces the expected parsed scores $\mu_\theta(x)$ toward the weak supervision label $s_l(x)$ in expectation over the training distribution. This result links the GRPO objective to stable scoring behavior and constitutes the training-side foundation of our end-to-end detection analysis.

While characterizing training-time alignment, inference introduces an additional uncertainty: each evaluation produces a random score $S_\theta(x)$. Proposition 1 provides Hoeffding's concentration bound for $K$ parsed scores.

**Proposition 1.** *(**Multi-sampling concentration**) Fix an inference input $x$. Suppose the $K$ well-formed generations $Y_1, \ldots, Y_K \overset{i.i.d.}{\sim} f_\theta(\cdot \mid x)$. For each $k \in \{1, \ldots, K\}$, define the parsed score $S_\theta^{(k)}(x) \triangleq s_p(x, Y_k) \in \{1, \ldots, 10\}$, and let $\bar{s}(x) \triangleq \frac{1}{K}\sum_{k=1}^{K} S_\theta^{(k)}(x)$ denote the aggregated score in Eqn. (6). Then for any bias threshold $u > 0$,*

$$\mathbb{P}\big(|\bar{s}(x) - \mathbb{E}[S_\theta(x) \mid x]| \geq u \mid x\big) \leq 2\exp\Big(-\frac{2Ku^2}{(10-1)^2}\Big). \qquad (8)$$

Due to the requirement of domain knowledge reserves or expert intervention, a rigorous human-level annotation of hallucination severity $s^\star$ is almost inaccessible. Accordingly,

we treat the similarity-based label $s_l$ as a weak proxy and model it as: $s_l(x) = g(s^\star(x)) + \epsilon(x)$, where $g$ is a monotone non-decreasing mapping, and $\epsilon$ captures conditional bias arising from stochasticity and systematic knowledge discrepancies. For the detection process based on this, we have the following derivations.

**Corollary 1.** *(**Ranking error decomposition**) Let $x$ denote an input and $s_l(x)$ its corresponding weak label. Assume the proxy bias is uniformly bounded: $|\epsilon(x)| \leq b_{\max}$, $\forall x$. Let $Y_1, \ldots, Y_K \overset{i.i.d.}{\sim} f_\theta(\cdot \mid x)$ be well-formed generations, and define $S_\theta^{(k)}(x)$ and $\bar{s}(x)$ as in Proposition 1. Let $x^+$ and $x^-$ denote independent draws from the distributions of true (non-hallucinated) and hallucinated inputs, respectively. Define the ranking error probability $\mathcal{E}_{\text{rank}} \triangleq \mathbb{P}\big(\bar{s}(x^+) \leq \bar{s}(x^-)\big)$. Then for any $\Delta > b_{\max}$,*

$$\mathcal{E}_{\text{rank}} \leq \underbrace{\mathbb{P}\big(g(s^\star(x^+)) - g(s^\star(x^-)) \leq 2\Delta\big)}_{\textit{intrinsic separability}} + \underbrace{\frac{4\,\mathcal{J}'(\theta)}{\Delta - b_{\max}}}_{\textit{training alignment}}$$

$$+ \underbrace{4\exp\Big(-\frac{2K}{(10-1)^2}\Big(\frac{\Delta - b_{\max}}{2}\Big)^2\Big)}_{\textit{multi-sampling concentration}},$$

*where $\mathcal{J}'(\theta)$ is affine-equivalent to the GRPO objective $\mathcal{J}(\theta)$ defined in Eqn. (1).*

Corollary 1 establishes a decomposable upper bound on the detector's ranking error probability. This bound makes explicit the principal factors that affect the detection performance. Beyond the **intrinsic separability of the data**, it crucially shows that a smaller **training alignment loss** $\mathcal{J}'(\theta)$ and a larger **inference sampling size** $K$ both contribute to a reduction of the error bound. This provides theoretical support for our design choices: optimizing the GRPO objective effectively aligns the agent's scores with the proxy supervision, while the multi-sampling aggregation strategy suppresses inference variance at an exponential rate. Consequently, the proposed training and inference components jointly ensure reliable detection. We refer readers to Appendix A for more details.

# 5. Experiments

## 5.1. Experimental Settings

**Datasets.** We conduct the experiments on four standard QA datasets: TriviaQA (Joshi et al., 2017), SciQ (Welbl et al., 2017), NQ Open (Kwiatkowski et al., 2019), and CoQA (Reddy et al., 2019). For NQ Open, we use all $3,610$ samples from the validation set. From TriviaQA, we randomly sample $3,310$ test pairs while $3,000$ training pairs are drawn from SciQ and CoQA. Each dataset is split into training and test sets with a $3:1$ ratio, and all reported results are averaged over $5$ independent random splits. All

*Table 2.* Comparisons with hallucination detection baselines on different datasets for LLaMA-3.1-8b and Qwen-3-8b in terms of AUROC (%), where ♣ denotes methods trained on fully labeled datasets.

| Model | Method | TriviaQA | SciQ | NQ Open | CoQA | Avg. |
|---|---|---|---|---|---|---|
| Llama-3.1-8b | LN-Entropy (Malinin & Gales, 2021) | $73.62_{\pm2.20}$ | $62.69_{\pm2.73}$ | $52.36_{\pm1.53}$ | $74.52_{\pm1.86}$ | 65.80 |
| | Self-evaluation (Kadavath et al., 2022) | $56.07_{\pm1.92}$ | $54.12_{\pm1.82}$ | $59.83_{\pm3.68}$ | $62.51_{\pm1.42}$ | 58.13 |
| | CCS (Burns et al., 2022) | $78.20_{\pm1.89}$ | $58.85_{\pm3.03}$ | $55.50_{\pm1.89}$ | $68.98_{\pm2.04}$ | 65.38 |
| | SelfCKGPT (Manakul et al., 2023) | $74.58_{\pm1.90}$ | $59.68_{\pm2.45}$ | $62.13_{\pm2.60}$ | $70.61_{\pm2.10}$ | 66.75 |
| | Perplexity (Ren et al., 2023) | $80.62_{\pm2.62}$ | $66.12_{\pm2.85}$ | $57.92_{\pm2.60}$ | $81.41_{\pm2.21}$ | 71.52 |
| | SAPLMA♣(Azaria & Mitchell, 2023) | $78.51_{\pm3.16}$ | $85.63_{\pm0.96}$ | $76.23_{\pm0.82}$ | $71.58_{\pm1.35}$ | 77.99 |
| | Semantic Entropy (Kuhn et al., 2023) | $78.71_{\pm3.09}$ | $77.81_{\pm3.17}$ | $61.04_{\pm4.29}$ | $75.26_{\pm4.63}$ | 73.21 |
| | Lexical Similarity (Lin et al., 2024) | $77.96_{\pm2.03}$ | $67.09_{\pm2.05}$ | $62.85_{\pm1.57}$ | $77.53_{\pm0.90}$ | 71.36 |
| | EigenScore (Chen et al., 2024a) | $51.35_{\pm1.23}$ | $51.52_{\pm0.82}$ | $52.17_{\pm2.13}$ | $52.00_{\pm1.19}$ | 51.76 |
| | HaloScope♣ (Du et al., 2024) | $58.19_{\pm5.79}$ | $69.04_{\pm6.36}$ | $63.38_{\pm3.02}$ | $72.11_{\pm4.96}$ | 65.68 |
| | TAD ♣ (Vazhentsev et al., 2025) | $72.01_{\pm1.03}$ | $66.75_{\pm1.96}$ | $88.88_{\pm2.56}$ | $74.86_{\pm2.10}$ | 70.63 |
| | TSV♣ (Park et al., 2025) | $79.78_{\pm3.36}$ | $80.01_{\pm1.17}$ | $70.17_{\pm1.41}$ | $69.31_{\pm6.75}$ | 74.82 |
| | **HCPD (Ours)** | $\mathbf{86.25}_{\pm1.08}$ | $\mathbf{86.04}_{\pm2.25}$ | $\mathbf{90.38}_{\pm3.58}$ | $\mathbf{90.07}_{\pm2.58}$ | **88.19** |
| Qwen-3-8b | LN-Entropy (Malinin & Gales, 2021) | $64.66_{\pm2.55}$ | $61.22_{\pm4.44}$ | $68.70_{\pm4.11}$ | $54.28_{\pm3.51}$ | 62.22 |
| | Self-evaluation (Kadavath et al., 2022) | $54.74_{\pm3.59}$ | $51.36_{\pm0.36}$ | $60.02_{\pm1.18}$ | $57.01_{\pm1.44}$ | 55.78 |
| | CCS (Burns et al., 2022) | $57.51_{\pm1.97}$ | $57.96_{\pm3.03}$ | $58.63_{\pm5.53}$ | $55.10_{\pm4.58}$ | 57.30 |
| | SelfCKGPT (Manakul et al., 2023) | $77.12_{\pm2.73}$ | $66.78_{\pm2.40}$ | $80.51_{\pm4.53}$ | $67.57_{\pm2.30}$ | 73.00 |
| | Perplexity (Ren et al., 2023) | $59.65_{\pm2.42}$ | $59.27_{\pm4.59}$ | $67.81_{\pm4.32}$ | $54.06_{\pm3.63}$ | 60.20 |
| | SAPLMA♣(Azaria & Mitchell, 2023) | $78.11_{\pm1.09}$ | $86.63_{\pm1.53}$ | $72.86_{\pm1.20}$ | $80.28_{\pm1.40}$ | 79.47 |
| | Semantic Entropy (Kuhn et al., 2023) | $81.74_{\pm2.78}$ | $70.84_{\pm5.41}$ | $75.10_{\pm1.97}$ | $70.59_{\pm5.09}$ | 74.57 |
| | Lexical Similarity (Lin et al., 2024) | $52.33_{\pm1.44}$ | $53.61_{\pm2.77}$ | $56.75_{\pm4.30}$ | $59.13_{\pm2.32}$ | 55.45 |
| | EigenScore (Chen et al., 2024a) | $52.60_{\pm2.03}$ | $52.85_{\pm2.57}$ | $56.39_{\pm3.26}$ | $52.79_{\pm0.37}$ | 53.66 |
| | HaloScope♣ (Du et al., 2024) | $58.21_{\pm3.99}$ | $74.98_{\pm3.98}$ | $57.25_{\pm1.50}$ | $62.18_{\pm4.49}$ | 63.16 |
| | TAD ♣ (Vazhentsev et al., 2025) | $71.27_{\pm1.90}$ | $55.98_{\pm4.37}$ | $76.82_{\pm3.77}$ | $66.94_{\pm0.58}$ | 67.75 |
| | TSV♣ (Park et al., 2025) | $73.42_{\pm3.89}$ | $78.77_{\pm0.94}$ | $61.38_{\pm3.43}$ | $68.40_{\pm4.92}$ | 70.49 |
| | **HCPD (Ours)** | $\mathbf{93.69}_{\pm0.89}$ | $\mathbf{92.63}_{\pm2.90}$ | $\mathbf{87.35}_{\pm6.22}$ | $\mathbf{84.80}_{\pm1.01}$ | **89.62** |

model responses are generated using greedy decoding. More details on dataset are shown in Appendix C.1.

**Baselines.** We compare our method against various baselines from multiple technical paradigms for hallucination detection. 1) Logit-based approaches: Perplexity (Ren et al., 2023), Length Normalized Entropy (LN-entropy) (Malinin & Gales, 2021) and Semantic Entropy (Kuhn et al., 2023); 2) Consistency-based approaches: Lexical Similarity (Lin et al., 2024), SelfCKGPT (Manakul et al., 2023) and EigenScore (Chen et al., 2024a); 3) Verbalized-confidence approach: Self-evaluation (Kadavath et al., 2022); 4) Internal state-based approaches: Contrast-Consistent Search (CCS) (Burns et al., 2022), SAPLMA (Azaria & Mitchell, 2023), HaloScope (Du et al., 2024), TAD (Vazhentsev et al., 2025) and TSV (Park et al., 2025).

**Implementation Details.** We compare our method with baselines on 7 LLMs, i.e., LLaMA-3.1-8b, LLaMA-3.1-70b (Dubey et al., 2024), LLaMA-2-7b, LLaMA-2-13b (Touvron et al., 2023b), Qwen-3-8b (Yang et al., 2025a), Qwen-2.5-7b, and Qwen-2.5-14b (Yang et al., 2024). For our method, we adopt a Qwen-2.5-7b as the agent and train it according to the Open-R1 implementation[4]. Through the optimization parameters in our method, we set $lr = 2 \times 10^{-4}$ on LLaMA-3.1-8b and $lr = 1 \times 10^{-4}$ on Qwen-3-8b. For the coefficient $\beta$ that controls the $D_{KL}$ strength, we set $\beta = 0.05$ on TriviaQA, SciQ with Qwen-3-8b and TriviaQA

with LLaMA-3.1-8b, while $\beta = 0.04$ for all others. More details are shown in Appendix C.2.

**Evaluation.** Following Farquhar et al. (2024), we evaluate detection performance using the Area Under the Receiver Operating Characteristic Curve (AUROC). As a proxy for ground-truth factuality, we employ BLEURT (Sellam et al., 2020) to measure semantic consistency between a generated answer and its reference. Generated answers with a consistency score exceeding 0.5 are regarded as relatively faithful, whereas those below are labeled as hallucinated.

### 5.2. Comparisons with Baselines

**Results on LLaMA-3.1-8b.** As reported in Table 2, most training-free methods exhibit limited detection performance (about 51.76% to 73.21% on average). While some methods display moderate separability on simple benchmarks (*e.g.*, TriviaQA, CoQA), their performance drops significantly on more challenging datasets. In contrast, training-based methods maintain stronger and more consistent performance across datasets, such as SAPLMA (77.99%) and TSV (74.82%). Notably, our method attains the best overall results while relying solely on $(q, a)$ inputs, with an average AUROC of 88.19%, which exceeds the second-best method by 10.20%. Specifically, HCPD improves AUROC by 5.63% on TriviaQA, by 14.15% on NQ Open, and by 8.66% on CoQA. We attribute these gains to the agent that fully considers criteria from multiple perspectives, enabling

[4]https://github.com/huggingface/open-r1

*Table 3.* Comparisons with training-based baselines across target models on TriviaQA in terms of AUROC (%).

| Source Model | Method | Target Model | | | | | | |
| --- | --- | --- | --- | --- | --- | --- | --- | --- |
| | | LLaMA-3.1-8b | LLaMA-3.1-70b | LLaMA-2-7b | LLaMA-2-13b | Qwen-3-8b | Qwen-2.5-7b | Qwen-2.5-14b |
| LLaMA-3.1-8b | SAPLMA (Azaria & Mitchell, 2023) | $78.51_{\pm3.16}$ | $77.63_{\pm2.63}$ | $78.13_{\pm2.65}$ | $78.25_{\pm1.83}$ | $71.63_{\pm2.15}$ | $69.70_{\pm2.13}$ | $63.77_{\pm2.28}$ |
| | HaloScope (Du et al., 2024) | $58.19_{\pm6.10}$ | $86.50_{\pm5.56}$ | $82.68_{\pm2.27}$ | $90.88_{\pm1.85}$ | $54.99_{\pm1.89}$ | $68.37_{\pm3.66}$ | $68.33_{\pm3.77}$ |
| | TSV (Park et al., 2025) | $79.78_{\pm3.36}$ | $81.29_{\pm10.23}$ | $82.85_{\pm3.90}$ | $88.06_{\pm4.23}$ | $59.89_{\pm8.38}$ | $77.07_{\pm5.82}$ | $61.17_{\pm7.98}$ |
| | **HCPD (Ours)** | $\mathbf{86.25}_{\pm1.08}$ | $\mathbf{86.87}_{\pm0.98}$ | $\mathbf{90.74}_{\pm0.52}$ | $\mathbf{93.43}_{\pm1.33}$ | $\mathbf{78.89}_{\pm2.22}$ | $\mathbf{88.84}_{\pm1.02}$ | $\mathbf{83.34}_{\pm0.86}$ |
| Qwen-3-8b | SAPLMA (Azaria & Mitchell, 2023) | $78.22_{\pm2.64}$ | $78.14_{\pm1.64}$ | $80.33_{\pm1.22}$ | $79.50_{\pm1.28}$ | $78.11_{\pm1.09}$ | $71.22_{\pm1.16}$ | $71.73_{\pm2.18}$ |
| | HaloScope (Du et al., 2024) | $53.89_{\pm0.78}$ | $57.97_{\pm1.21}$ | $55.47_{\pm9.50}$ | $56.73_{\pm9.89}$ | $58.21_{\pm3.99}$ | $57.64_{\pm1.53}$ | $78.65_{\pm5.95}$ |
| | TSV (Park et al., 2025) | $54.14_{\pm3.26}$ | $58.85_{\pm5.70}$ | $60.31_{\pm6.89}$ | $64.04_{\pm8.75}$ | $73.42_{\pm3.89}$ | $57.08_{\pm3.75}$ | $65.77_{\pm4.82}$ |
| | **HCPD (Ours)** | $\mathbf{87.09}_{\pm1.32}$ | $\mathbf{89.73}_{\pm2.63}$ | $\mathbf{90.84}_{\pm0.76}$ | $\mathbf{93.95}_{\pm1.07}$ | $\mathbf{93.69}_{\pm0.89}$ | $\mathbf{89.74}_{\pm0.77}$ | $\mathbf{87.94}_{\pm1.19}$ |

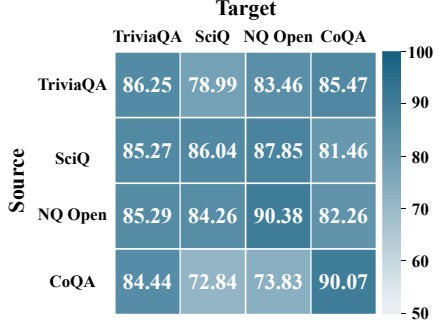

*Figure 2.* Cross-dataset AUROCs of HCPD on LLaMA-3.1-8b.

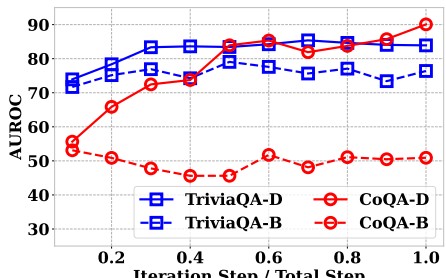

*Figure 3.* Impact of reward design, where "-D" denotes differentiable scoring reward and "-B" denotes binary scoring reward.

*Table 4.* Controlled empirical decomposition of HCPD.

| Method | AUROC |
| --- | --- |
| Self-evaluation (Kadavath et al., 2022) | $56.07_{\pm1.92}$ |
| HCPD (Ours, Pre-RL) | $66.54_{\pm0.59}$ |
| HCPD (Ours, Post-RL) | $86.25_{\pm1.08}$ |

*Table 5.* Impact of sampling size $K$.

| Dataset | 1 | 2 | 5 | 10 |
| --- | --- | --- | --- | --- |
| TriviaQA | $85.21_{\pm1.22}$ | $85.71_{\pm0.86}$ | $86.25_{\pm1.08}$ | $86.25_{\pm0.98}$ |
| SciQ | $85.06_{\pm2.33}$ | $85.47_{\pm2.12}$ | $86.04_{\pm2.25}$ | $86.14_{\pm1.89}$ |
| NQ Open | $86.89_{\pm1.60}$ | $89.26_{\pm2.46}$ | $90.38_{\pm3.58}$ | $90.41_{\pm2.46}$ |
| CoQA | $88.60_{\pm4.12}$ | $89.37_{\pm3.34}$ | $90.07_{\pm2.58}$ | $89.85_{\pm2.66}$ |
| Inf. Time (s) | 0.2349 | 0.4675 | 1.1313 | 2.2807 |

fine-grained identification of hallucinated content.

**Results on Qwen-3-8b.** The results on Qwen-3-8b exhibit similar characteristics: the training-free and training-based methods achieve average AUROCs of $53.66\%-74.57\%$ and $63.16\%-79.47\%$, respectively. In comparison, our HCPD again achieves consistent improvements outperforming the second-best method by $10.15\%$ on average. Taken together with the results on LLaMA-3.1-8b across datasets, the effectiveness and robustness of HCPD under diverse backbone models and benchmarks are significantly underscored.

**Transferability across target models.** A key advantage of HCPD is its *model-agnostic* design: it operates in the natural language space, which is universally shared across LLMs, thereby achieving superior transferability across diverse target models. To demonstrate the advantages of HCPD, we further compare it with other training-based baselines under a transfer setting where the detector is evaluated on outputs generated by target models from different families and scales. Results in Table 3 show that when applied to unseen target models, competing baselines such as HaloScope and TSV often suffer noticeable performance degradation due to distributional shifts in features of the proxy model. In contrast, HCPD maintains stable and excellent performance across heterogeneous target models, which undoubtedly better meet the needs of practical deployment in open, real-world settings. More results on the other 3 datasets are

shown in Appendix D.1.

**Transferability across data distributions.** Furthermore, the proposed Human-like Criteria Probing mechanism enables our method to assess queries with varying distributions and formats in a flexible manner. To evaluate distributional transfer, we test HCPD under cross-dataset settings. As shown in Figure 2, HCPD exhibits amazing generalization when transferred to TriviaQA, SciQ, and NQ Open, preserving high performance. We observe a modest degradation on CoQA, which we attribute to substantial differences in the underlying QA formulation and interaction mode. Additional results on Qwen-3-8b are shown in Appendix D.2.

### 5.3. Ablation Studies

**Controlled empirical decomposition.** To isolate the contributions of our proposed human-like criteria probing mechanism and weakly-supervised alignment training, we conducted a controlled empirical decomposition using standard Self-evaluation (Kadavath et al., 2022) as a baseline. As detailed in Table 4, HCPD outperforms the baselines by Human-like Criteria Probing itself, yielding an AUROC improvement of $10.47\%$. This performance is further augmented by our GRPO alignment framework, which delivers a substantial subsequent AUROC gain of $19.71\%$.

*Table 6.* Comparisons with baselines under alternative metrics on TriviaQA for LLaMA-3.1-8b.

| Method | Metric | |
|---|---|---|
| | ROUGE | DeepSeek |
| LN-Entropy (Malinin & Gales, 2021) | $77.70_{\pm 1.39}$ | $52.69_{\pm 1.19}$ |
| CCS (Burns et al., 2022) | $82.92_{\pm 1.64}$ | $52.52_{\pm 2.12}$ |
| SelfCKGPT (Manakul et al., 2023) | $73.92_{\pm 1.26}$ | $71.38_{\pm 0.90}$ |
| Perplexity (Ren et al., 2023) | $85.49_{\pm 1.13}$ | $53.82_{\pm 2.86}$ |
| SAPLMA (Azaria & Mitchell, 2023) | $87.22_{\pm 1.13}$ | $79.99_{\pm 0.92}$ |
| Lexical Similarity (Lin et al., 2024) | $82.40_{\pm 0.90}$ | $53.63_{\pm 1.69}$ |
| HaloScope (Du et al., 2024) | $72.99_{\pm 4.61}$ | $57.98_{\pm 4.96}$ |
| TSV (Park et al., 2025) | $78.46_{\pm 6.15}$ | $73.31_{\pm 5.63}$ |
| **HCPD (Ours)** | $\mathbf{89.17}_{\pm 0.42}$ | $\mathbf{80.42}_{\pm 0.39}$ |

**Impact of reward design.** We discussed the advantages of differentiable scoring over binary scoring in Section 4.4, and here we further validate this claim empirically. As shown in Figure 3, replacing the graded reward with a binary "0/1" signal not only downgrades the evaluation to binary classification, but also yields a significant reduction in performance (from 86.25% to 79.06% on TriviaQA and from 90.07% to 51.75% on CoQA). We attribute this degradation to the lack of severity information, which makes the agent more prone to misclassifying samples near the decision threshold.

**Impact of sampling size.** According to the Corollary 1, the ranking error decreases as the number of samples $K$ increases. The results in Table 5 provide quantitative support for this trend. Larger $K$ yields progressively improved accuracy and inference stability, but meanwhile incurs a linear increase in computational cost. To balance performance and efficiency, we set $K = 5$ in experiments. Additional performance and consumption comparison with baselines are detailed in Appendix D.4. Notably, HCPD matches the speed of light metrics and outpaces consistency-based methods. Moreover, its model-agnostic design yields greater computational savings when evaluating outputs from large target models (*e.g.*, LLaMA-2-13B, Qwen-2.5-14B, and LLaMA-3.1-70B).

**Beyond the BLEURT metric.** It is notable that BLEURT is an option, not a dependency. We conducte GRPO using alternative signals (*e.g.*, ROUGE (Lin et al., 2024), and DeepSeek-V3 as a Judge (Liu et al., 2024)). Results in Table 6 confirm that as an evaluation metric provided, HCPD seamlessly aligns its behavior to the reward signal and achieves strong performance. The detailed implementation of DeepSeek-V3 is shown in Appendix C.2.

## 6. Conclusion

In this paper, we propose *Human-like Criteria Probing* (HCP), an interpretable paradigm that achieves hallucination detection under the zero-source setting. We instantiate HCP-Detector as an LLM-based scoring agent that decomposes evaluation into context-relevant criteria, assigns adaptive weights, and produces evidence-grounded analyses along with an overall score. The agent is further aligned via GRPO using a dense score-alignment reward derived from weak proxy supervision, which is based on semantic consistency signals on QA benchmarks. It employs multi-sampling aggregation at inference time to suppress stochastic variance and yield stable predictions. Both theoretical analysis and extensive experiments across diverse datasets and model architectures demonstrate the effectiveness of our HCPD in identifying hallucinated content.

## Acknowledgments

This work was partially supported by the Joint Funds of the National Natural Science Foundation of China (Grant No.U24A20327), Key-Area Research and Development Program Guangdong Province 2018B010107001, and TCL Science and Technology Innovation Fund, China. Jiahao Yang is supported by the China Scholarship Council (CSC) under Grant No. 202506150018.

## Impact Statement

This work proposes Human-like Criteria Probing for Hallucination Detection (HCPD), a zero-source framework for detecting factual and logical inconsistencies in LLM outputs. By operating solely on the observed query–response pair, HCPD is intended to improve reliability and transparency in practical black-box deployment settings, with particular relevance to safety-critical domains such as healthcare and education. Furthermore, HCPD provides inherent explainability by outputting the specific criteria and weights used for each decision, which can aid developers in model debugging and offer end-users transparent justifications. We believe this research contributes positively to the development of more accountable and transparent AI systems.

## Author Contributions

Jiahao Yang, Shuhai Zhang, and Hailong Kang contributed equally to this work. Feng Liu, Qi Chen, and Mingkui Tan are the corresponding authors. Jiahao Yang and Shuhai Zhang conceived the main idea and designed the proposed method. Jiahao Yang and Hailong Kang conducted the experiments and performed the main empirical analysis. Jiahao Yang, Shuhai Zhang, and Hailong Kang contributed to manuscript writing, result organization, and paper revision. Feng Liu, Qi Chen, and Mingkui Tan supervised the project, provided guidance on methodology and experiments, and revised the manuscript.

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

# APPENDIX

## Contents

# A. Theoretical Analysis

## A.1. Setup and Notations

Let $x = (q, a)$ be an input of hallucination detection under the zero-source constraint, where $a$ can be generated by any source LLM. The scoring agent $f_\theta$ produces a structured and interpretable evaluation whose predicted score is an integer random variable $s_p(x) \in \{1, \ldots, 10\}$ in the well-formed regime. During training, each query $q$ has a ground-truth reference answer $\hat{a}$ within dataset, and an auxiliary LLM generates candidate answers $\{a^{(n)}\}_{n=1}^N$. Weak supervision labels $s_l^{(n)} \in \{1, \ldots, 10\}$ are computed from a semantic consistency metric $\text{sim}(\hat{a}, a^{(n)})$ (based on BLEURT (Sellam et al., 2020)). The agent is optimized with GRPO using the score-alignment reward given by Eqn. (5). During inference, we invoke the agent $K$ times and aggregate $\bar{s}$ according to Eqn. (6).

## A.2. Proof of Theorem 1

**Theorem 1** (*Expectation alignment under training*) *Let $x$ denote an input and $s_l(x)$ its corresponding weak label. Consider the $\ell_1$-risk of predicting score $s$:*

$$R_x(s) \triangleq |s - s_l(x)|,$$

*whose minimizer is $s = s_l(x)$. Let $Y \sim f_\theta(\cdot \mid x)$ be a stochastic well-formed generation of the agent, and let $S_\theta(x) = s_p(x, Y)$ denote the parsed score. Define the conditional mean score:*

$$\mu_\theta(x) \triangleq \mathbb{E}\big[S_\theta(x) \mid x\big].$$

*Then we have:*

$$\mathbb{E}_x\big[|\mu_\theta(x) - s_l(x)|\big] \leq \mathcal{J}'(\theta),$$

*where the expectation is taken over the training distribution of $x$, and $\mathcal{J}'(\theta)$ is affine-equivalent to the GRPO objective $\mathcal{J}(\theta)$ defined in Eqn. (1).*

*Proof.* Given a fixed input $x$, since $s_l(x)$ is deterministic, we have

$$|\mu_\theta(x) - s_l(x)| = |\mathbb{E}\big[S_\theta(x) \mid x\big] - s_l(x)| = |\mathbb{E}\big[S_\theta(x) - s_l(x) \mid x\big]|.$$

By Jensen's inequality (Jensen, 1906),

$$|\mathbb{E}\big[S_\theta(x) - s_l(x) \mid x\big]| \leq \mathbb{E}\big[|S_\theta(x) - s_l(x)| \mid x\big].$$

Taking expectation over the training distribution of $x$ yields

$$\mathbb{E}_x\big[|\mu_\theta(x) - s_l(x)|\big] \leq \mathbb{E}_x\mathbb{E}\big[|S_\theta(x) - s_l(x)| \mid x\big] = \mathbb{E}_{x,Y}\big[|s_p(x, Y) - s_l(x)|\big]. \tag{9}$$

By the score-alignment reward in Eqn. (5)), minimizing the KL-regularized GRPO objective in Eqn. (1) is equivalent up to affine constants to minimizing the nonnegative objective

$$\begin{aligned}
\mathcal{J}'(\theta) &= \mathbb{E}_x\mathbb{E}_{Y \sim f_\theta(\cdot|x)}\big[|s_p(x, Y) - s_l(x)|\big] + \lambda \cdot \mathbb{E}_x D_{KL}\big(f_\theta(\cdot|x)||f_0(\cdot|x)\big) \\
&= \mathbb{E}_{x,Y}\big[|s_p(x, Y) - s_l(x)|\big] + \lambda \cdot \mathbb{E}_x D_{KL}\big(f_\theta(\cdot|x)||f_0(\cdot|x)\big).
\end{aligned} \tag{10}$$

where $\lambda \propto \beta$. Since the non-negativity of the KL regularizer, we have

$$\mathbb{E}_{x,Y}\big[|s_p(x, Y) - s_l(x)|\big] \leq \mathcal{J}'(\theta).$$

Combining the two inequalities Eqn. (9) and Eqn. (10) completes the proof. □

## A.3. Proof of Corollary 1

**Proposition 1** (*Multi-sampling concentration*) *Fix an inference input $x$. Suppose the $K$ well-formed generations $Y_1, \ldots, Y_K \overset{i.i.d.}{\sim} f_\theta(\cdot \mid x)$. For each $k \in \{1, \ldots, K\}$, define the parsed score*

$$S_\theta^{(k)}(x) \triangleq s_p(x, Y_k) \in \{1, \ldots, 10\},$$

*and let $\bar{s}(x) \triangleq \frac{1}{K} \sum_{k=1}^K S_\theta^{(k)}(x)$ denote the aggregated score in Eqn. (6). Then for any bias threshold $u > 0$,*

$$\mathbb{P}\big(|\bar{s}(x) - \mathbb{E}[S_\theta(x) \mid x]| \geq u \mid x\big) \leq 2 \exp\Big(-\frac{2Ku^2}{(10-1)^2}\Big).$$

*Proof.* Conditional on the input $x$, the random variables $\{S_\theta^{(k)}(x)\}_{k=1}^K$ are *i.i.d* and each are bounded in $[1, 10]$. Let $\mu_\theta(x) \triangleq \mathbb{E}\big[S_\theta(x) \mid x\big]$. By Hoeffding's inequality for bounded *i.i.d* variables, we have

$$\mathbb{P}\big(|\bar{s}(x) - \mu_\theta(x)| \geq u \mid x\big) \leq 2 \exp\Big(-\frac{2Ku^2}{(10-1)^2}\Big).$$

which is exactly the claim. $\qquad\square$

**Corollary 1** (*Ranking error decomposition*) *Let $x$ denote an input and $s_l(x)$ its corresponding weak label. Assume the proxy bias is uniformly bounded:*

$$|\epsilon(x)| \leq b_{\max}, \quad \forall x.$$

*Let $Y_1, \ldots, Y_K \overset{i.i.d.}{\sim} f_\theta(\cdot \mid x)$ be well-formed generations, and define $S_\theta^{(k)}(x)$ and $\bar{s}(x)$ as in Proposition 1. Let $x^+$ and $x^-$ denote independent draws from the distributions of true (non-hallucinated) and hallucinated inputs, respectively. Define the ranking error probability:*

$$\mathcal{E}_{\mathrm{rank}} \triangleq \mathbb{P}\big(\bar{s}(x^+) \leq \bar{s}(x^-)\big).$$

*Then for any $\Delta > b_{\max}$,*

$$\mathcal{E}_{\mathrm{rank}} \leq \underbrace{\mathbb{P}\big(g(s^\star(x^+)) - g(s^\star(x^-)) \leq 2\Delta\big)}_{\text{intrinsic separability}} + \underbrace{\frac{4\,\mathcal{J}'(\theta)}{\Delta - b_{\max}}}_{\text{training alignment}} + \underbrace{4 \exp\Big(-\frac{2K}{(10-1)^2}\Big(\frac{\Delta - b_{\max}}{2}\Big)^2\Big)}_{\text{multi-sampling concentration}},$$

*where $\mathcal{J}'(\theta)$ is affine-equivalent to the GRPO objective $\mathcal{J}(\theta)$ defined in Eqn. (1).*

*Proof.* Let $\Delta > b_{\max}$ and define $\delta \triangleq \frac{\Delta - b_{\max}}{2} > 0$. Consider the event

$$A \triangleq \{g(s^\star(x^+)) - g(s^\star(x^-)) \leq 2\Delta\}.$$

On the complement $A^c$, we have $g(s^\star(x^+)) - g(s^\star(x^-)) \geq 2\Delta$.

If in addition scores are close to their truthfulness scores $|\bar{s}(x) - g(s^\star(x))| \leq \Delta$, then

$$\bar{s}(x^+) \geq g(s^\star(x^+)) - \Delta \geq g(s^\star(x^-)) + \Delta \geq \bar{s}(x^-)$$

which means order-preserving. Therefore, the ranking error event $\{\bar{s}(x^+) \leq \bar{s}(x^-)\}$ is contained in

$$A \cup \{|\bar{s}(x^+) - g(s^\star(x^+))| > \Delta\} \cup \{|\bar{s}(x^-) - g(s^\star(x^-))| > \Delta\}$$

Taking probability and using the union bound gives

$$\mathcal{E}_{\mathrm{rank}} \leq \mathbb{P}(A) + \mathbb{P}(|\bar{s}(x^+) - g(s^\star(x^+))| > \Delta) + \mathbb{P}(|\bar{s}(x^-) - g(s^\star(x^-))| > \Delta). \qquad (11)$$

For a fixed input $x$, the proxy bias satisfies $|\epsilon(x)| \leq b_{\max}$, so

$$|\bar{s}(x) - g(s^\star(x))| \leq |\bar{s}(x) - s_l(x)| + |s_l(x) - g(s^\star(x))| \leq |\bar{s}(x) - s_l(x)| + b_{\max}.$$

Thus $|\bar{s}(x) - g(s^\star(x))| > \Delta$ implies $|\bar{s}(x) - s_l(x)| \geq \Delta - b_{\max} = 2\delta$.

Meanwhile, we have

$$|\bar{s}(x) - s_l(x)| \leq |\bar{s}(x) - \mu_\theta(x)| + |\mu_\theta(x) - s_l(x)|,$$

hence $|\bar{s}(x) - s_l(x)| > 2\delta$ implies

$$|\bar{s}(x) - \mu_\theta(x)| > \delta \text{ or } |\mu_\theta(x) - s_l(x)| > \delta,$$

so again by union bound, we have

$$\mathbb{P}(|\bar{s}(x) - g(s^\star(x))| > \Delta) \leq \mathbb{P}(|\bar{s}(x) - \mu_\theta(x)| > \delta) + \mathbb{P}(|\mu_\theta(x) - s_l(x)| > \delta). \tag{12}$$

Conditional on $x$, Proposition 1 yields

$$\mathbb{P}\big(|\bar{s}(x) - \mu_\theta(x)| \geq \delta \mid x\big) \leq 2\exp\Big(-\frac{2K\delta^2}{(10-1)^2}\Big).$$

Taking expectation over $x$ preserves the same upper bound:

$$\mathbb{P}\big(|\bar{s}(x) - \mu_\theta(x)| \geq \delta\big) \leq 2\exp\Big(-\frac{2K\delta^2}{(10-1)^2}\Big). \tag{13}$$

Moreover, by Markov's inequality (Ross, 2020),

$$\mathbb{P}(|\mu_\theta(x) - s_l(x)| > \delta) \leq \frac{\mathbb{E}_x\big[|\mu_\theta(x) - s_l(x)|\big]}{\delta}.$$

Using Theorem 1, we have $\mathbb{E}_x\big[|\mu_\theta(x) - s_l(x)|\big] \leq \mathcal{J}'(\theta)$, hence

$$\mathbb{P}(|\mu_\theta(x) - s_l(x)| > \delta) \leq \frac{\mathcal{J}'(\theta)}{\delta}. \tag{14}$$

By combining Eqn. (12)-(14), we have for any $\Delta > b_{\max}$ and $\delta = \frac{\Delta - b_{\max}}{2}$,

$$\mathbb{P}(|\bar{s}(x) - g(s^\star(x))| > \Delta) \leq \frac{\mathcal{J}'(\theta)}{\delta} + 2\exp\Big(-\frac{2K\delta^2}{(10-1)^2}\Big). \tag{15}$$

Apply Eqn. (15) to $x^+$ and $x^-$ in Eqn. (11), we have

$$
\begin{aligned}
\mathcal{E}_{\mathrm{rank}} &\leq \mathbb{P}(A) + \mathbb{P}(|\bar{s}(x^+) - g(s^\star(x^+))| > \Delta) + \mathbb{P}(|\bar{s}(x^-) - g(s^\star(x^-))| > \Delta) \\
&= \mathbb{P}(A) + 2\Big(\frac{\mathcal{J}'(\theta)}{\delta} + 2\exp\big(-\frac{2K\delta^2}{(10-1)^2}\big)\Big) \\
&= \mathbb{P}\big(g(s^\star(x^+)) - g(s^\star(x^-)) \leq 2\Delta\big) + \frac{4\,\mathcal{J}'(\theta)}{\Delta - b_{\max}} + 4\exp\Big(-\frac{2K}{(10-1)^2}\Big(\frac{\Delta - b_{\max}}{2}\Big)^2\Big),
\end{aligned}
\tag{16}
$$

by recalling $\delta = \frac{\Delta - b_{\max}}{2}$, which is exactly the desired decomposition. $\square$

# B. More Related Work

**Large Language Models (LLMs).** LLMs have become central to modern natural language processing, demonstrating strong capabilities in reasoning (Niu et al., 2022; Wang et al., 2023; 2024; Chen et al., 2025; Zhang et al., 2025), knowledge grounding (Li et al., 2020; Guo et al., 2025), and multi-modal understanding (Zhao et al., 2024; Ji et al., 2024). Among open-source ecosystems, two influential model families are LLaMA (Touvron et al., 2023a;b; Dubey et al., 2024; Grattafiori et al., 2024) and Qwen (Bai et al., 2023; Team, 2024; Yang et al., 2024; 2025a).

The **LLaMA** series, developed by Meta, has been instrumental in advancing open-weight LLMs. LLaMA (Touvron et al., 2023a) demonstrated that competitive models can be trained on publicly available corpora, incorporating architectural components such as RMSNorm (Zhang & Sennrich, 2019), SwiGLU activations (Shazeer, 2020), and Rotary Positional Embeddings (RoPE). LLaMA 2 (Touvron et al., 2023b) scaled pre-training data, introduced RLHF-tuned conversational variants, and adopted a more permissive license. LLaMA 3 (Dubey et al., 2024) further scaled model capacity (up to 70B parameters) and multilingual data, together with an improved tokenizer and stronger instruction-following performance. The most recent iteration, LLaMA 3.1 (Grattafiori et al., 2024), extends the context window to 128k tokens via enhanced RoPE scaling, enlarges the pre-training corpus to over 15 trillion tokens, and integrates advanced post-training techniques such as large-scale Direct Preference Optimization (DPO) (Rafailov et al., 2023) and emergent tool-use capabilities, achieving state-of-the-art results among open models and approaching parity with leading proprietary systems.

The **Qwen** series, first released in 2023 (Bai et al., 2023), emphasizes scalability and practical deployment, offering models across a wide range of parameter sizes. Qwen-1.5[5] and Qwen-2 (Team, 2024) extend context length to 128k tokens across the family and incorporate Grouped Query Attention (GQA) (Ainslie et al., 2023) to improve inference efficiency, leading to substantial performance gains. Qwen2.5 (Yang et al., 2024) further introduces domain-specialized variants for coding and mathematics, forming a flexible generalist–specialist design. The most recent Qwen3 (Yang et al., 2025a) proposes a Hybrid-Reasoning paradigm that dynamically switches between a deliberate *Thinking* mode for complex reasoning and a lightweight *Non-Thinking* mode for rapid responses, enabling explicit trade-offs between computational cost and accuracy. Its large-scale Mixture-of-Experts (MoE) (Shazeer et al., 2017; Zhuang et al., 2018) variants achieve competitive state-of-the-art performance, while smaller dense models retain strong parameter efficiency.

**Reinforcement Learning.** Reinforcement learning (RL) has demonstrated effectiveness in improving the reasoning ability of LLMs by aligning their behaviors with human preferences or task-specific objectives (Ouyang et al., 2022).

Similar to conventional machine learning, Supervised Fine-Tuning (SFT) trains LLMs by maximizing the log-likelihood on high-quality, task-specific demonstration data. While SFT effectively transfers knowledge, it is inherently constrained by the quality and coverage of the demonstrations and cannot directly optimize non-differentiable or long-horizon objectives. Serving as the subsequent step, Proximal Policy Optimization (PPO) (Schulman et al., 2017) introduces a reward model trained from human preference comparisons and maximizes this reward while constraining policy updates to maintain stability. This PPO-based Reinforcement Learning from Human Feedback (RLHF) (Ouyang et al., 2022) enables models like InstructGPT (Ouyang et al., 2022) and ChatGPT[6], substantially improving instruction-following and helpfulness. However, it is computationally intensive and operationally complex, with the requirement of joint training and balancing of a policy model, a reward model, and a value function. Direct Preference Optimization (DPO) (Rafailov et al., 2023) emerged as a simplified and more stable alternative, bypassing explicit reward modeling by deriving a closed-form objective that directly optimizes the policy using pairwise preference data. By reformulating the RL objective as a supervised loss, DPO significantly reduces training complexity compared to PPO. Nonetheless, its reliance on the Bradley–Terry (Hunter, 2004; Zeng et al., 2019) preference model makes it less amenable to scenarios involving dense, scalar reward signals, such as the score-alignment supervision adopted in this work. Group Relative Policy Optimization (GRPO) (Shao et al., 2024) is a recent advancement designed to improve stability and efficiency in LLM alignment. It operates by sampling a group of candidate outputs for each prompt, assigning rewards for each, and optimizing the policy by increasing the relative likelihood of higher-reward outputs within the group. This group-relative ranking objective provides a stable and efficient learning signal, making GRPO particularly well-suited for weakly supervised, scalar reward settings.

---

[5]https://qwenlm.github.io/blog/qwen1.5/
[6]https://openai.com/blog/chatgpt

# C. More Details for Experiment Settings

## C.1. More Details on Datasets

**TriviaQA** (Joshi et al., 2017) is a large-scale reading comprehension benchmark comprising $95,956$ question-answer pairs, each supplemented with an average of approximately 6 supporting evidence documents, yielding $662,659$ associated documents in total sourced from Wikipedia and the Web. Since questions are authored independently of the subsequent evidence retrieval, the dataset exhibits substantial lexical and syntactic divergence between queries and supporting evidence, and frequently requires multi-sentence reasoning to answer accurately.

**SciQ** (Welbl et al., 2017) is a science question answering benchmark designed to evaluate domain knowledge and reasoning. It comprises $13,679$ manually collected scientific exam questions spanning physics, chemistry, biology, and related subjects, split into $11,679$ training, $1,000$ validation, and $1,000$ test instances. Each question is multiple-choice with 4 answer options, and most are accompanied by an additional paragraph that supports the correct answer. SciQ also provides a direct-answer variant in which distractors are removed to enable reading-comprehension-style evaluation.

**NQ Open** (Kwiatkowski et al., 2019) is an open-domain question answering benchmark derived from Natural Questions corpus, comprising anonymized real-world Google queries paired with brief answers, all of which can be answered through content from the English Wikipedia. The dataset contains $91,535$ question–answer pairs, including $87,925$ training instances and $3,610$ validation instances. To ensure concise and standardized responses, examples with answer lengths exceeding five words are discarded, yielding a high-quality benchmark for open-domain QA evaluation.

**CoQA** (Reddy et al., 2019) is a large-scale benchmark for conversational question answering, which aims to measure a model's ability to understand a given passage and answer a series of interdependent questions that appear in a conversation. The dataset contains over $127,000$ questions with answers collected from more than $8,000$ conversations. Each conversation is collected by pairing two crowd workers to interact over a passage through multi-turn question and answer exchanges. CoQA captures challenging phenomena that are not present in standard reading comprehension benchmarks, including coreference and pragmatic reasoning.

**Wikipedia**[7] is a large-scale multilingual dataset comprising cleaned articles from Wikipedia dumps across $320+$ languages. It includes $61.6$ million+ rows of text, with each entry containing an article's ID, URL, title, and markdown-stripped content. The dataset supports tasks like text generation and masked-language modeling, with one subset per language and a single training split, making it a foundational resource for multilingual NLP research.

## C.2. More Details on Implementation

### C.2.1. IMPLEMENTATION DETAILS ON BASELINES.

We implement baselines spanning 4 paradigms, largely following the official implementations when available.

**(1) Logit-based methods.** We implement *Perplexity* (Ren et al., 2023) using the public codebase[8], which adopts sequence-level perplexity as the detection score; and *LN-Entropy* (Malinin & Gales, 2021) based on the same codebase[8], which computes entropy with length normalization; and *Semantic Entropy* (Kuhn et al., 2023) using the official repository[9], which clusters semantically equivalent generations before entropy estimation.

**(2) Consistency-based methods.** We implement *Lexical Similarity* (Lin et al., 2024) following the codebase[8], using ROUGE-based similarity to measure agreement across multiple sampled responses; and *SelfCheckGPT* (Manakul et al., 2023) using the same codebase[8], which aggregates multiple similarity metrics (*e.g.*, BERTScore) for self-consistency evaluation; and *EigenScore* (Chen et al., 2024a) following the codebase[8], which evaluates semantic consistency via eigenvalue statistics of the response covariance matrix in embedding space.

**(3) Verbalized-confidence methods.** We implement *Self-evaluation* (Kadavath et al., 2022) following the codebase[9], which prompts the model to estimate the probability that its answer is correct.

**(4) Internal-state methods.** We implement *CCS* (Burns et al., 2022) using the official codebase[10], which extracts latent

---

[7] https://dumps.wikimedia.org
[8] https://github.com/D2I-ai/eigenscore
[9] https://github.com/jlko/semantic_uncertainty
[10] https://github.com/collin-burns/discovering_latent_knowledge

knowledge signals from model activations; and *SAPLMA* (Azaria & Mitchell, 2023) with the provided repository[11], which trains auxiliary classifiers on hidden representations; and *HaloScope* (Du et al., 2024) using the official implementation[12], which applies SVD to identify hallucination-relevant subspaces; and *TSV* (Park et al., 2025) following the codebase[13], which learns steering vectors to reshape latent features for improved separability.

### C.2.2. IMPLEMENTATION DETAILS ON OUR METHOD.

In our main experiments, we instantiate the agent $f_\theta$ with Qwen-2.5-7b (Yang et al., 2024) to assess the generated answer $a$ conditioned on the query $q$. To elicit expert-like behavior and implement Human-like Criteria Probing, we follow the Generalist Reward Model design in SPCT (Liu et al., 2025) and format each input pair $(q, a)$ using the template below:

---

**Hallucination detection prompt template (PART I)**

You are a skilled little expert at detecting hallucinations in AI responses. You should evaluate given responses based on the given judging criteria.

Given the context of the conversation (the last round is the User's query) and multiple responses from the Assistant, you need to refer to the [General Evaluation Criteria] to score the responses for **hallucination risk**. Based on the general evaluation criteria, state potential other specific criteria to the query, the weights of different criteria, and then provide an overall comprehensive score upon them.

Each score is an integer between 1 and 10, where a **higher score indicates lower hallucination risk and higher reliability** (i.e., the response is more factually grounded, logically sound, and appropriate). For example:
- Score 1: Severe hallucination; largely fabricated or misleading
- Score 6: Moderate reliability; contains noticeable but non-critical hallucinations
- Score 10: No hallucination; fully accurate, consistent, and appropriate

#### General Evaluation Criteria ####
1. Factual Verification
  - Extract all factual claims and validate against scientific consensus, historical records, or authoritative sources.
  - Fully Grounded (9–10 points): All claims are verifiable, accurate, and aligned with authoritative knowledge.
  - Mostly Accurate (6–8 points): Minor inaccuracies (e.g., slight statistic deviation <5%, or non-critical outdated info), but core facts are correct.
  - Partially Grounded (3–5 points): Noticeable factual errors (e.g., wrong person/event, >5% statistical error, or mention of fictional entities as real).
  - Severely Fabricated (1–2 points): Multiple false claims, completely invented narratives, or reliance on non-existent sources.

2. Logical Consistency
  - Map argument dependencies and flag contradictions or reasoning flaws.
  - Fully Consistent (9–10 points): Reasoning is sound, premises support conclusions, no contradictions.
  - Generally Consistent (6–8 points): Minor logical gaps or weak inferences, but overall argument holds.
  - Partially Consistent (3–5 points): Clear issues such as self-contradiction, circular reasoning, or unsupported conclusions.
  - Critically Inconsistent (1–2 points): Multiple contradictions, paradoxical statements, or reasoning that collapses under basic scrutiny.

3. Semantic Accuracy
  - Identify term misuse, ambiguity, or information loss that distorts meaning.

---

The prompt specifies a *General Evaluation Criteria* set $\mathcal{C}$ that includes *"Factual Verification"*, *"Logical Consistency"*, *"Semantic Accuracy"*, *"Social Fairness"*, and *"Timeline Verification"*, together with their associated scoring guidelines. It

---

[11] https://github.com/ivanrozhd/anlp-project
[12] https://github.com/deeplearning-wisc/haloscope
[13] https://github.com/deeplearning-wisc/tsv

further enforces a strictly structured and interpretable output schema to facilitate downstream aggregation.

Notably, we employ distinct sampling strategies (temperature $T$, nucleus sampling parameter top-$p$, and repetition penalty $RP$) for the training and test phases to serve different objectives. In particular, we set $T = $ top-$p = RP = 1.0$ to endow $f_\theta$ with greater flexibility to explore the evaluation space more broadly and learn robust assessment behaviors; and we adopt a lower-entropy (more deterministic) setting with $T = 0.9$, top-$p = 0.9$, $RP = 1.1$ to obtain higher-confidence and more stable evaluations in inference.

---

**Hallucination detection prompt template (PART II)**

- Precise & Clear (9–10 points): Terms used correctly; meaning is unambiguous and fully preserved.
- Mostly Clear (6–8 points): Occasional vague phrasing or mild over-abstraction, but core message remains intact.
- Partially Accurate (3–5 points): Misused technical terms, misleading analogies, or significant information loss affecting understanding.
- Highly Distorted (1–2 points): Language so ambiguous, abstract, or erroneous that it conveys false or incomprehensible meaning.

4. Social Fairness
- Detect stereotypes, discrimination, or culturally insensitive content.
- Fully Inclusive (9–10 points): Language is respectful, neutral, culturally aware, and free of bias.
- Generally Appropriate (6–8 points): Minor generalizations or slightly outdated phrasing, but not offensive.
- Partially Biased (3–5 points): Clear stereotypes, gender/region/culture-based assumptions, or insensitive framing.
- Highly Offensive (1–2 points): Promotes discrimination, harmful generalizations, or dehumanizing language.

5. Timeline Verification
- Annotate time-related claims and validate temporal logic and recency.
- Temporally Sound (9–10 points): All time references are accurate, logically ordered, and appropriately current.
- Mostly Accurate (6–8 points): Minor recency issues (e.g., citing 6-year-old data in a stable domain) or slightly vague timeline.
- Partially Accurate (3–5 points): Confused chronology, treating future speculation as fact, or using obsolete info (>5 years) in critical contexts.
- Severely Anachronistic (1–2 points): Asserts future events as past, creates impossible timelines, or mixes historical eras illogically (e.g., "Einstein used smartphones").

#### Conversation Context ####
{conversation context & query}

#### Responses to be Scored ####
[The Begin of Response i]
{the i-th response}
[The End of Response i]

#### Output Format Requirements ####
Specific Criteria: <Other criteria and weights>
Analysis: <Comparison based on criteria>
Scores: \boxed{x, x} (for multiple responses)

---

For reward-based alignment with GRPO (Shao et al., 2024), we adopt the Open-R1 implementation[14] and conduct training on 2 NVIDIA A800 GPUs. We use a per-device batch size of 10. Through the optimization parameters in RL, the learning rate is set to $lr = 2 \times 10^{-4}$ on LLaMA-3.1-8b for all 4 datasets, while $lr = 1 \times 10^{-4}$ on Qwen-3-8b. For the coefficient $\beta$ that controls the strength of $D_{KL}$, we set $\beta = 0.05$ on TriviaQA with LLaMA-3.1-8b and Qwen-3-8b as well as SciQ with Qwen-3-8b, while $\beta = 0.04$ for all others. All reported results are averaged over 5 independent random splits.

---

[14]https://github.com/huggingface/open-r1

C.2.3. TRAINING DATA CONSTRUCTION AND LABELING.

Following Kuhn et al. (2023), we construct hallucination detection datasets from standard question-answer (QA) benchmarks. Specifically, for traditional QA datasets that do not require additional reference context (*e.g.*, TriviaQA (Joshi et al., 2017), SciQ (Welbl et al., 2017), and NQ Open (Kwiatkowski et al., 2019)), we generate candidate answers for each question $q$ using the following prompt template:

> **Prompt template for closed-source QA dataset**
>
> Answer the question concisely:
> Q: {question}
> A:

For open-source QA datasets with auxiliary context (*e.g.*, CoQA (Reddy et al., 2019)), we adopt the following prompt:

> **Prompt template for open-source QA dataset**
>
> Based on the following context, answer the question concisely:
> Context: {context}
> Q: {question}
> A:

We also employ different sampling strategies for the training and test splits. In the **training split**, we aim to generate a set of responses $\{a^{(n)}\}_{n=1}^N$ that spans a broad spectrum from factually correct to clearly hallucinated, providing fine-grained supervision for reinforcement learning. We therefore adopt a higher-entropy sampling strategy with $T = 0.5, \text{top-}p = 1.0, RP = 1.0$, and set $N = 9$, yielding 10 answers per query in total when including the ground-truth reference $\hat{a}$ for training. In the **test split**, we generate a single response per query using greedy decoding with 5 beam search, matching the evaluation protocol in Kuhn et al. (2023). Notably, although this procedure may appear superficially similar to the agent setting, it differs fundamentally in both its target (the generator rather than the evaluation agent) and its purpose (producing diverse candidate answers rather than learning reliable scoring behavior).

For labeling, we assign the reference answer $\hat{a}$ the maximum score $s_l = 10$ and adopt BLEURT (Sellam et al., 2020) to score the generated answers. The resulting graded score is derived from Eqn. (4). For binary evaluation on the test set, we further threshold the BLEURT-based similarity: answers with the similarity above $0.5$ are labeled as *True*, and those below as *Hallucinated*. It should be emphasized that the reference answer $\hat{a}$ is only used offline to construct labels during training and never accessed at inference; at test time, the detector takes only $(q, a)$ where $a$ can be produced by any LLM.

In ablation experiments that using alternative signals, we adopt Deepseek-V3 (Liu et al., 2024) to evaluate the truthfulness of generated content, following the LLM-as-a-judge (Zheng et al., 2023) paradigm. Specifically, we assess the truthfulness of LLM-generated responses by verifying their semantic equivalence to the provided gold standard answers. The input prompt for this evaluation on QA datasets is as:

> **Prompt template for QA dataset**
>
> You are tasked with evaluating whether a generated answer is truthful compared to a provided list of gold standard answers. Use a 10-point scoring scale (1-2: Completely inconsistent/hallucinated, 3-4: Mostly inconsistent with minor, 5-6: Partially consistent, mixed quality, 7-8: Mostly consistent with minor errors, 9-10: Completely consistent, accurate, comprehensive) to score the generated content. Provide a lear justification for your scoring.
>
> Response Format:
>  - Score: [1/2/3/4/5/6/7/8/9/10]
>  - Justification: [Explain briefly why the answer is correct or incorrect.]
>
> Question: {question}
> Gold Standard Answers: {reference_answers}
> Generated Answer: {generated_answer}

And the input prompt Wikipedia dataset is as:

---

**Prompt template for Wikipedia dataset**

You are tasked with evaluating whether a generated Wikipedia passage is consistent with the provided real Wikipedia passage. Use a 10-point scoring scale (1-2: Completely inconsistent/hallucinated, 3-4: Mostly inconsistent with minor, 5-6: Partially consistent, mixed quality, 7-8: Mostly consistent with minor errors, 9-10: Completely consistent, accurate, comprehensive) to score the generated content. Provide a clear justification for your scoring.
Response Format:
 - Score: [1/2/3/4/5/6/7/8/9/10]
 - Justification: [Explain briefly why the answer is correct or incorrect.]

Title: {question}
Real Wikipedia passage: {reference_answers}
Generated Wikipedia passage: {generated_answer}

---

### C.2.4. PSEUDO CODE OF HCPD

---

**Algorithm 1** Reward-based Alignment Training of HCPD

---

**Input:** Dataset $\mathcal{D} = \{(q_i, \{(a_i^{(n)}, s_i^{(n)})\})\}$, group size $G$, general criteria set $\mathcal{C}$, initial agent $f_\theta \leftarrow f_0, \eta$;

Form detection input $x_i^{(n)} \leftarrow (q_i, a_i^{(n)})$;

**for** $x_i^{(n)}$ in $\mathcal{D}$ **do**

    Sample a group of evaluations $\{Y_g\}_{g=1}^G \sim f_\theta(\cdot \mid x; \mathcal{C})$;

    **for** $g = 1, 2, \ldots, G$ **do**

        Parse predicted score $s_p^{(g)} \leftarrow s_p(x, Y_g) \in \{1, \ldots, 10\}$;

        Compute reward $r_g$ via Eqn. (5);

        Compute advantages $A_g \leftarrow r_g - \frac{1}{G}\sum_{j=1}^G r_j$;

    **end for**

    Compute GRPO objective via Eqn. (1);

    $\theta \leftarrow \theta + \eta\nabla_\theta \mathcal{J}_x(\theta)$.

**end for**

**Output:** $f_\theta^*$

---

**Algorithm 2** Detection via Multi-sampling Aggregation

---

**Input:** Scoring agent $f_\theta^*$, general criteria set $\mathcal{C}$, input $x = (q, a)$, $K$;

Sample $K$ evaluations $\{Y_k\}_{k=1}^K \sim f_\theta^*(\cdot \mid x; \mathcal{C})$;

**for** $k = 1, 2, \ldots, K$ **do**

    Parse score $s_p^{(k)} \leftarrow s_p(x, Y_k) \in \{1, \ldots, 10\}$;

**end for**

Aggregate scores via Eqn. (6);

**Output:** Truthfulness Score $\bar{s}$

---

# D. More Experiment Results

## D.1. More Results on Transferability across Target Models

To further demonstrate the advantages of HCPD's *model-agnostic* design, we provide cross-target evaluation on the remaining 3 datasets. Results in Table 7-9 show that our method maintains consistently strong performance across heterogeneous target models. Notably, when faced with earlier-generation models (*e.g.*, LLaMA-2 and Qwen-2.5), detection performance further improves, likely because such models produce less fluent and less realistic outputs, making hallucinations easier to distinguish. Overall, these results indicate that HCPD is undoubtedly well-suited for detecting hallucinations from complex input sources, which is common in real-world deployment.

*Table 7.* Comparisons with training-based baselines across target models on SciQ in terms of AUROC (%).

| Source Model | Method | Target Model | | | | | |
|---|---|---|---|---|---|---|---|
| | | LLaMA-3.1-8b | LLaMA-2-7b | LLaMA-2-13b | Qwen-3-8b | Qwen-2.5-7b | Qwen-2.5-14b |
| LLaMA-3.1-8b | SAPLMA (Azaria & Mitchell, 2023) | $85.63_{\pm0.96}$ | $79.77_{\pm2.45}$ | $70.96_{\pm2.64}$ | $79.05_{\pm2.61}$ | $74.65_{\pm1.88}$ | $63.16_{\pm1.83}$ |
| | HaloScope (Du et al., 2024) | $69.04_{\pm6.36}$ | $75.64_{\pm9.33}$ | $78.95_{\pm8.82}$ | $62.87_{\pm10.34}$ | $66.15_{\pm11.19}$ | $61.65_{\pm8.32}$ |
| | TSV (Park et al., 2025) | $80.01_{\pm1.17}$ | $68.77_{\pm10.12}$ | $61.86_{\pm7.32}$ | $70.46_{\pm9.86}$ | $64.56_{\pm7.16}$ | $56.10_{\pm2.27}$ |
| | **HCPD (Ours)** | $\mathbf{86.04}_{\pm2.25}$ | $\mathbf{92.47}_{\pm1.44}$ | $\mathbf{95.48}_{\pm1.26}$ | $\mathbf{91.89}_{\pm1.07}$ | $\mathbf{89.92}_{\pm2.08}$ | $\mathbf{78.20}_{\pm7.42}$ |
| Qwen-3-8b | SAPLMA (Azaria & Mitchell, 2023) | $85.21_{\pm1.71}$ | $84.11_{\pm0.93}$ | $79.88_{\pm1.65}$ | $86.63_{\pm1.53}$ | $77.08_{\pm1.66}$ | $68.39_{\pm3.73}$ |
| | HaloScope (Du et al., 2024) | $53.60_{\pm4.11}$ | $58.31_{\pm8.59}$ | $57.42_{\pm7.30}$ | $74.98_{\pm4.19}$ | $65.18_{\pm10.56}$ | $59.16_{\pm7.85}$ |
| | TSV (Park et al., 2025) | $63.67_{\pm3.91}$ | $57.88_{\pm6.40}$ | $56.89_{\pm5.85}$ | $78.77_{\pm0.94}$ | $60.67_{\pm2.00}$ | $59.06_{\pm3.88}$ |
| | **HCPD (Ours)** | $\mathbf{85.39}_{\pm2.70}$ | $\mathbf{90.51}_{\pm1.28}$ | $\mathbf{96.63}_{\pm0.86}$ | $\mathbf{92.63}_{\pm2.90}$ | $\mathbf{92.57}_{\pm3.14}$ | $\mathbf{84.21}_{\pm4.75}$ |

*Table 8.* Comparisons with training-based baselines across target models on NQ Open in terms of AUROC (%).

| Source Model | Method | Target Model | | | | | |
|---|---|---|---|---|---|---|---|
| | | LLaMA-3.1-8b | LLaMA-2-7b | LLaMA-2-13b | Qwen-3-8b | Qwen-2.5-7b | Qwen-2.5-14b |
| LLaMA-3.1-8b | SAPLMA (Azaria & Mitchell, 2023) | $76.23_{\pm0.82}$ | $66.77_{\pm1.43}$ | $65.55_{\pm2.04}$ | $68.27_{\pm2.34}$ | $63.18_{\pm1.53}$ | $63.82_{\pm1.43}$ |
| | HaloScope (Du et al., 2024) | $63.38_{\pm3.02}$ | $74.35_{\pm9.93}$ | $62.58_{\pm7.66}$ | $63.68_{\pm11.04}$ | $59.10_{\pm7.32}$ | $60.66_{\pm4.97}$ |
| | TSV (Park et al., 2025) | $70.17_{\pm1.47}$ | $58.42_{\pm3.87}$ | $58.20_{\pm3.31}$ | $61.00_{\pm1.57}$ | $60.43_{\pm3.68}$ | $57.24_{\pm2.98}$ |
| | **HCPD (Ours)** | $\mathbf{90.38}_{\pm3.58}$ | $\mathbf{92.94}_{\pm0.75}$ | $\mathbf{90.42}_{\pm1.92}$ | $\mathbf{92.45}_{\pm1.77}$ | $\mathbf{85.88}_{\pm6.03}$ | $\mathbf{78.62}_{\pm6.69}$ |
| Qwen-3-8b | SAPLMA (Azaria & Mitchell, 2023) | $70.55_{\pm3.11}$ | $66.85_{\pm0.67}$ | $61.85_{\pm1.50}$ | $72.86_{\pm1.20}$ | $68.52_{\pm1.95}$ | $\mathbf{68.87}_{\pm2.20}$ |
| | HaloScope (Du et al., 2024) | $62.83_{\pm9.95}$ | $64.84_{\pm12.85}$ | $55.49_{\pm4.94}$ | $57.25_{\pm1.50}$ | $62.00_{\pm7.79}$ | $58.60_{\pm6.65}$ |
| | TSV (Park et al., 2025) | $53.94_{\pm4.57}$ | $57.30_{\pm3.92}$ | $56.10_{\pm3.84}$ | $61.38_{\pm3.43}$ | $57.91_{\pm3.73}$ | $55.56_{\pm2.53}$ |
| | **HCPD (Ours)** | $\mathbf{79.42}_{\pm3.52}$ | $\mathbf{91.60}_{\pm1.41}$ | $\mathbf{93.04}_{\pm0.98}$ | $\mathbf{87.35}_{\pm6.22}$ | $\mathbf{85.70}_{\pm1.58}$ | $60.09_{\pm5.91}$ |

*Table 9.* Comparisons with training-based baselines across target models on CoQA in terms of AUROC (%).

| Source Model | Method | Target Model | | | | | |
|---|---|---|---|---|---|---|---|
| | | LLaMA-3.1-8b | LLaMA-2-7b | LLaMA-2-13b | Qwen-3-8b | Qwen-2.5-7b | Qwen-2.5-14b |
| LLaMA-3.1-8b | SAPLMA (Azaria & Mitchell, 2023) | $71.58_{\pm1.35}$ | $76.81_{\pm1.18}$ | $72.62_{\pm0.62}$ | $69.07_{\pm3.24}$ | $68.11_{\pm1.32}$ | $59.84_{\pm1.42}$ |
| | HaloScope (Du et al., 2024) | $72.11_{\pm4.96}$ | $58.10_{\pm6.07}$ | $62.53_{\pm7.26}$ | $57.75_{\pm2.67}$ | $67.05_{\pm6.23}$ | $53.53_{\pm2.05}$ |
| | TSV (Park et al., 2025) | $69.31_{\pm6.75}$ | $53.59_{\pm3.55}$ | $56.85_{\pm3.20}$ | $56.82_{\pm4.28}$ | $59.10_{\pm3.59}$ | $52.91_{\pm1.90}$ |
| | **HCPD (Ours)** | $\mathbf{90.07}_{\pm2.58}$ | $\mathbf{89.12}_{\pm1.24}$ | $\mathbf{89.38}_{\pm2.02}$ | $\mathbf{76.43}_{\pm3.03}$ | $\mathbf{84.19}_{\pm3.39}$ | $\mathbf{60.76}_{\pm6.24}$ |
| Qwen-3-8b | SAPLMA (Azaria & Mitchell, 2023) | $69.63_{\pm1.55}$ | $81.52_{\pm1.13}$ | $77.99_{\pm1.40}$ | $80.28_{\pm1.40}$ | $72.91_{\pm2.42}$ | $73.00_{\pm1.40}$ |
| | HaloScope (Du et al., 2024) | $74.46_{\pm8.04}$ | $65.96_{\pm12.78}$ | $69.45_{\pm7.78}$ | $62.18_{\pm4.49}$ | $64.07_{\pm10.56}$ | $54.10_{\pm2.21}$ |
| | TSV (Park et al., 2025) | $61.22_{\pm10.15}$ | $58.82_{\pm7.59}$ | $59.81_{\pm7.30}$ | $68.40_{\pm4.92}$ | $58.63_{\pm7.01}$ | $56.79_{\pm4.45}$ |
| | **HCPD (Ours)** | $\mathbf{88.43}_{\pm3.50}$ | $\mathbf{86.12}_{\pm0.83}$ | $\mathbf{88.09}_{\pm1.61}$ | $\mathbf{84.80}_{\pm1.01}$ | $\mathbf{83.14}_{\pm0.52}$ | $\mathbf{73.26}_{\pm1.25}$ |

### D.2. More Results on Transferability across Data Distributions

We also evaluate cross-dataset evaluation transfer on Qwen-3-8b. As shown in Figure 4, HCPD exhibits strong generalization under distribution shift, similar to that on LLaMA-3.1-8b. This indicates that by adaptively proposing criteria that fit the problem domain and format, HCPD is widely applicable across diverse hallucination detection scenarios. Notably, the agent trained on CoQA even outperforms its in-domain counterpart when transferred to the other three datasets. We hypothesize that CoQA's contextual setting provides richer supervision, which may improve knowledge coverage and contextual reasoning, thereby enhancing the agent's discriminative capability.

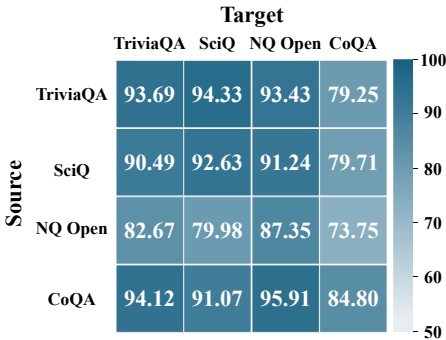

*Figure 4.* Cross-dataset AUROCs of HCPD on Qwen-3-8b.

### D.3. Impact of Generation Strategy

As mentioned above, we use different sampling strategies for the training and test phases to serve different objectives. To seek the most effective strategies for inference-time generation, we evaluate 5 sampling strategies with an increasing degree of freedom. Intuitively, higher freedom degree can broaden exploration across diverse perspectives, but it also reduces controllability and may yield unstable or less reliable evaluations (*e.g.*, $T = 1.3$, top-$p = 0.98$, $RP = 1.0$); conversely, overly deterministic sampling can constrain the assessment, potentially biasing it toward a single criterion or reasoning pattern (*e.g.*, $T = 0.5$, top-$p = 0.8$, $RP = 1.3$). Results in Table 10 indicate that HCPD can maintain excellent reasoning ability across strategies with diverse degree of freedom, which further supports the effectiveness of our alignment procedure.

*Table 10.* Impact of generation strategy.

| Dataset | Strategy ($T$ \| top-$p$ \| $RP$) | | | | |
| --- | --- | --- | --- | --- | --- |
| | 0.5 \| 0.8 \| 1.3 | 0.7 \| 0.85 \| 1.2 | 0.9 \| 0.9 \| 1.1 | 1.1 \| 0.95 \| 1.05 | 1.3 \| 0.98 \| 1.0 |
| TriviaQA | $85.51_{\pm1.38}$ | $85.90_{\pm1.44}$ | $\mathbf{86.25}_{\pm1.08}$ | $86.03_{\pm1.36}$ | $82.80_{\pm2.01}$ |
| SciQ | $85.59_{\pm0.44}$ | $\mathbf{86.62}_{\pm2.03}$ | $86.04_{\pm2.25}$ | $85.94_{\pm1.95}$ | $85.36_{\pm2.48}$ |
| NQ Open | $88.94_{\pm2.23}$ | $90.09_{\pm2.13}$ | $\mathbf{90.38}_{\pm3.58}$ | $90.08_{\pm2.17}$ | $87.30_{\pm3.00}$ |
| CoQA | $89.44_{\pm2.98}$ | $89.83_{\pm2.76}$ | $\mathbf{90.07}_{\pm2.58}$ | $89.85_{\pm2.67}$ | $88.14_{\pm4.04}$ |

### D.4. Performance and Consumption Comparison

In light of the practical deployment of HCPD, we analyze its computational overhead from 2 perspectives: *i) Inference time:* As shown in Table 5, the multi-sampling aggregation yields substantial performance gains at the expense of additional computation. However, $K = 5$ is not strictly necessary, since HCPD already outperforms all baselines at $K = 1$, reducing latency to $0.2349$s per sample; *ii) Inference VRAM:* Since most methods require LLM to extract internal states or probabilities, HCPD does not consume additional VRAM during inference. Moreover, its fixed 7B footprint is more efficient when evaluating large-scale LLMs. A detailed performance–consumption comparison is provided in Table 11, demonstrating that HCPD attains efficiency comparable to lightweight metrics while surpassing consistency-based methods.

### D.5. Generalization to Long-form Generation Task

Beyond factual hallucinations in short-form QA settings, we further evaluate the detection of faithful hallucinations in a Wikipedia article continuation task. To assess the generalization capability of HCPD in long-form generation, we directly apply the agent trained on TriviaQA to a Wikipedia continuation dataset. Notably, we employ DeepSeek-V3 as the evaluation metric (Liu et al., 2024), since most continuation passages exceed the 512-token limit of BLEURT. Detailed descriptions of

the dataset and prompting strategy are provided in Appendix C.1 and C.2, respectively. As shown in Table 12, As shown in Table 12, HCPD consistently outperforms existing baselines, demonstrating robust scalability to long-form generation tasks.

*Table 11.* Comparisons with baselines in terms of performance, inference time and GPU memory.

| Method | AUROC (%) ↑ | Inf. Time (s) ↓ | GPU Mem. (MiB) ↓ |
|---|---|---|---|
| LN-Entropy (Malinin & Gales, 2021) | $73.62_{\pm2.20}$ | 0.1383 | 17185 |
| CCS (Burns et al., 2022) | $78.20_{\pm1.89}$ | 1.1600 | 31498 |
| SelfCKGPT (Manakul et al., 2023) | $74.58_{\pm1.90}$ | 6.5310 | 17540 |
| Perplexity (Ren et al., 2023) | $80.62_{\pm2.62}$ | 0.1349 | 17185 |
| SAPLMA (Azaria & Mitchell, 2023) | $78.51_{\pm3.16}$ | 0.5341 | 16019 |
| Semantic Entropy (Kuhn et al., 2023) | $78.71_{\pm3.09}$ | 14.3026 | 17185 |
| Lexical Similarity (Lin et al., 2024) | $77.96_{\pm2.03}$ | 36.5646 | 17185 |
| EigenScore (Chen et al., 2024a) | $51.35_{\pm1.23}$ | 37.7550 | 17453 |
| HaloScope (Du et al., 2024) | $58.19_{\pm5.79}$ | 0.6012 | 16757 |
| TSV (Park et al., 2025) | $79.78_{\pm3.36}$ | 0.0934 | 23592 |
| HCPD (Ours, $K=1$) | $85.21_{\pm1.22}$ | 0.2349 | 18513 |
| HCPD (Ours, $K=5$) | $86.25_{\pm1.08}$ | 1.1313 | 19051 |

*Table 12.* Cross-dataset transfer performance on Wikipedia, using TriviaQA as the source dataset for training-based baselines.

| Method | AUROC |
|---|---|
| LN-Entropy (Malinin & Gales, 2021) | $73.02_{\pm1.76}$ |
| CCS (Burns et al., 2022) | $70.20_{\pm2.28}$ |
| SelfCKGPT (Manakul et al., 2023) | $72.37_{\pm2.03}$ |
| Perplexity (Ren et al., 2023) | $74.19_{\pm2.07}$ |
| HaloScope (Du et al., 2024) | $71.74_{\pm1.30}$ |
| TSV (Park et al., 2025) | $65.07_{\pm1.27}$ |
| **HCPD (Ours)** | $\mathbf{74.36}_{\pm1.37}$ |

# E. Future Directions

While HCPD provides an effective zero-source hallucination detector, several promising extensions warrant further investigation. Future work includes leveraging HCPD as a reward model within LLM training to explicitly suppress hallucinations, extending the framework to multimodal generation (*e.g.*, image-text or video-text) by incorporating modality-aware evaluation criteria, and generalizing human-like criteria probing mechanism to broader evaluation tasks such as safety, helpfulness, or style compliance. Furthermore, scaling HCPD to long-form generation, multi-turn dialogue, and tool-augmented systems will likely require hierarchical or claim-level scoring, as well as dialogue-aware consistency modeling. Collectively, these directions illustrate a principled path toward establishing HCP as a general, interpretable evaluation paradigm for increasingly complex and multimodal generation settings.

# F. Examples

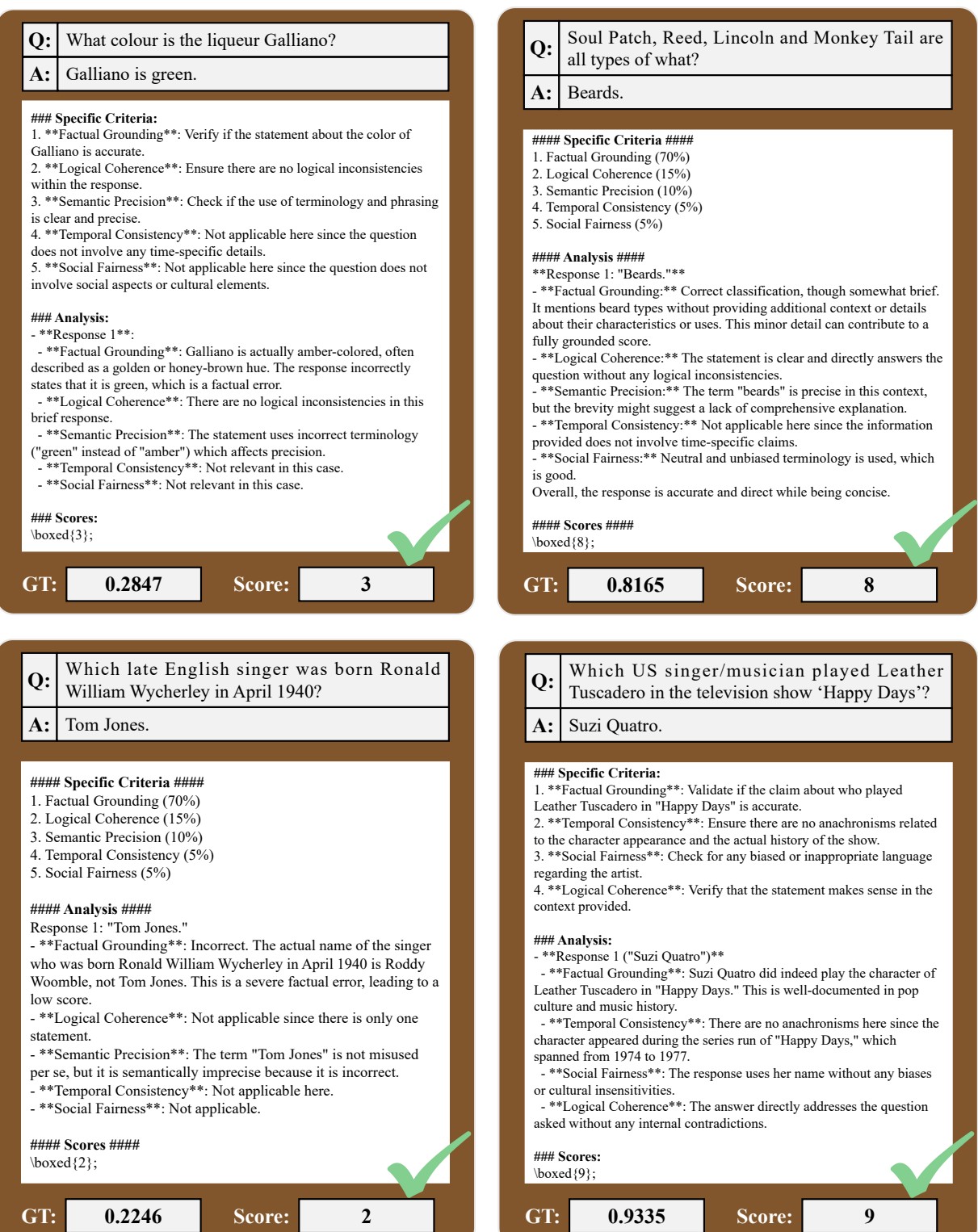

**Q:** What colour is the liqueur Galliano?

**A:** Galliano is green.

### Specific Criteria:
1. **Factual Grounding**: Verify if the statement about the color of Galliano is accurate.
2. **Logical Coherence**: Ensure there are no logical inconsistencies within the response.
3. **Semantic Precision**: Check if the use of terminology and phrasing is clear and precise.
4. **Temporal Consistency**: Not applicable here since the question does not involve any time-specific details.
5. **Social Fairness**: Not applicable here since the question does not involve social aspects or cultural elements.

### Analysis:
- **Response 1**:
  - **Factual Grounding**: Galliano is actually amber-colored, often described as a golden or honey-brown hue. The response incorrectly states that it is green, which is a factual error.
  - **Logical Coherence**: There are no logical inconsistencies in this brief response.
  - **Semantic Precision**: The statement uses incorrect terminology ("green" instead of "amber") which affects precision.
  - **Temporal Consistency**: Not relevant in this case.
  - **Social Fairness**: Not relevant in this case.

### Scores:
\boxed{3};

**GT:** 0.2847 **Score:** 3

---

**Q:** Soul Patch, Reed, Lincoln and Monkey Tail are all types of what?

**A:** Beards.

#### Specific Criteria ####
1. Factual Grounding (70%)
2. Logical Coherence (15%)
3. Semantic Precision (10%)
4. Temporal Consistency (5%)
5. Social Fairness (5%)

#### Analysis ####
**Response 1: "Beards."**
- **Factual Grounding:** Correct classification, though somewhat brief. It mentions beard types without providing additional context or details about their characteristics or uses. This minor detail can contribute to a fully grounded score.
- **Logical Coherence:** The statement is clear and directly answers the question without any logical inconsistencies.
- **Semantic Precision:** The term "beards" is precise in this context, but the brevity might suggest a lack of comprehensive explanation.
- **Temporal Consistency:** Not applicable here since the information provided does not involve time-specific claims.
- **Social Fairness:** Neutral and unbiased terminology is used, which is good.
Overall, the response is accurate and direct while being concise.

#### Scores ####
\boxed{8};

**GT:** 0.8165 **Score:** 8

---

**Q:** Which late English singer was born Ronald William Wycherley in April 1940?

**A:** Tom Jones.

#### Specific Criteria ####
1. Factual Grounding (70%)
2. Logical Coherence (15%)
3. Semantic Precision (10%)
4. Temporal Consistency (5%)
5. Social Fairness (5%)

#### Analysis ####
Response 1: "Tom Jones."
- **Factual Grounding**: Incorrect. The actual name of the singer who was born Ronald William Wycherley in April 1940 is Roddy Woomble, not Tom Jones. This is a severe factual error, leading to a low score.
- **Logical Coherence**: Not applicable since there is only one statement.
- **Semantic Precision**: The term "Tom Jones" is not misused per se, but it is semantically imprecise because it is incorrect.
- **Temporal Consistency**: Not applicable here.
- **Social Fairness**: Not applicable.

#### Scores ####
\boxed{2};

**GT:** 0.2246 **Score:** 2

---

**Q:** Which US singer/musician played Leather Tuscadero in the television show 'Happy Days'?

**A:** Suzi Quatro.

### Specific Criteria:
1. **Factual Grounding**: Validate if the claim about who played Leather Tuscadero in "Happy Days" is accurate.
2. **Temporal Consistency**: Ensure there are no anachronisms related to the character appearance and the actual history of the show.
3. **Social Fairness**: Check for any biased or inappropriate language regarding the artist.
4. **Logical Coherence**: Verify that the statement makes sense in the context provided.

### Analysis:
- **Response 1 ("Suzi Quatro")**
  - **Factual Grounding**: Suzi Quatro did indeed play the character of Leather Tuscadero in "Happy Days." This is well-documented in pop culture and music history.
  - **Temporal Consistency**: There are no anachronisms here since the character appeared during the series run of "Happy Days," which spanned from 1974 to 1977.
  - **Social Fairness**: The response uses her name without any biases or cultural insensitivities.
  - **Logical Coherence**: The answer directly addresses the question asked without any internal contradictions.

### Scores:
\boxed{9};

**GT:** 0.9335 **Score:** 9

*Figure 5.* Visualizations of successful detections via HCPD on TriviaQA.

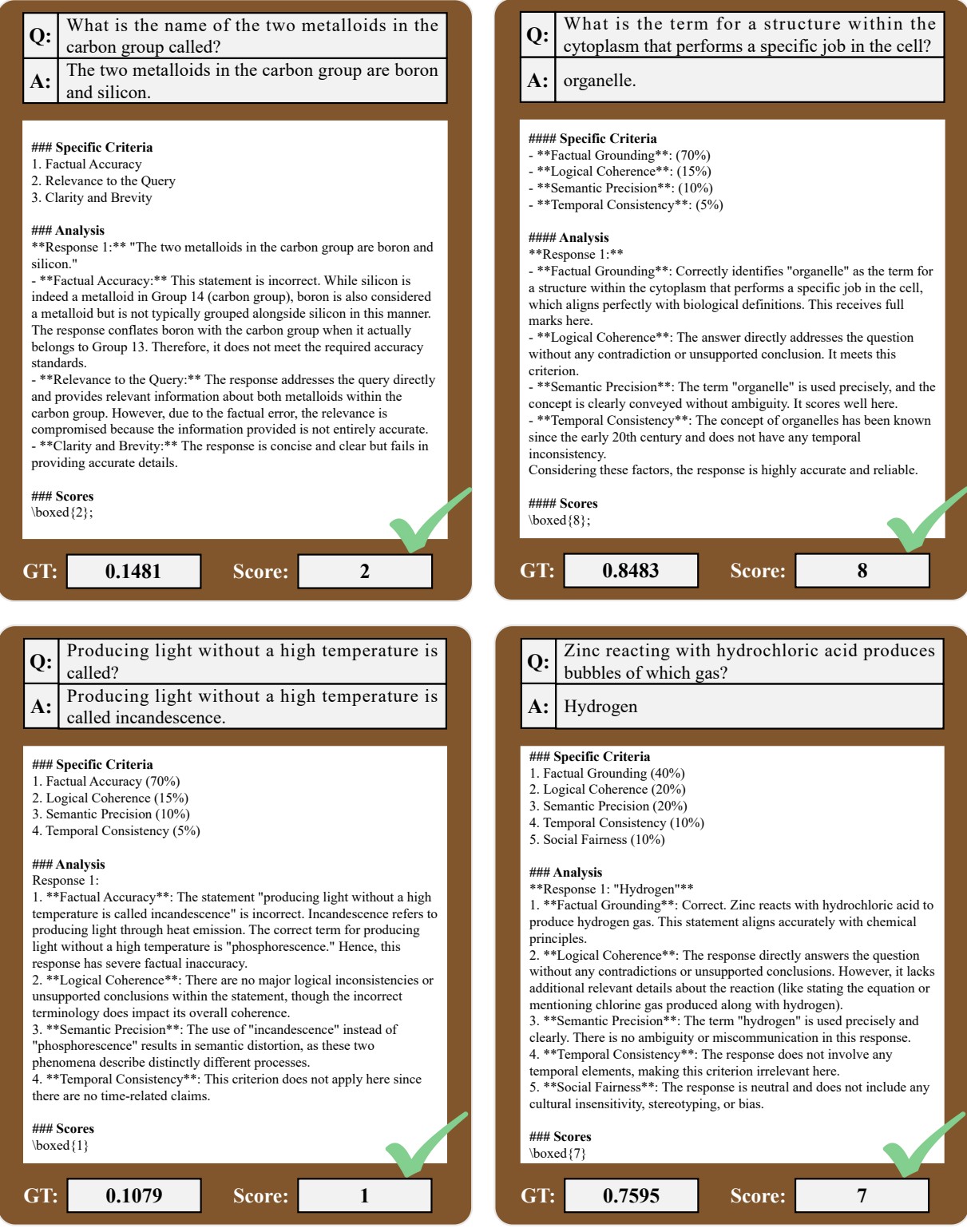

*Figure 6.* Visualizations of successful detections via HCPD on SciQ.

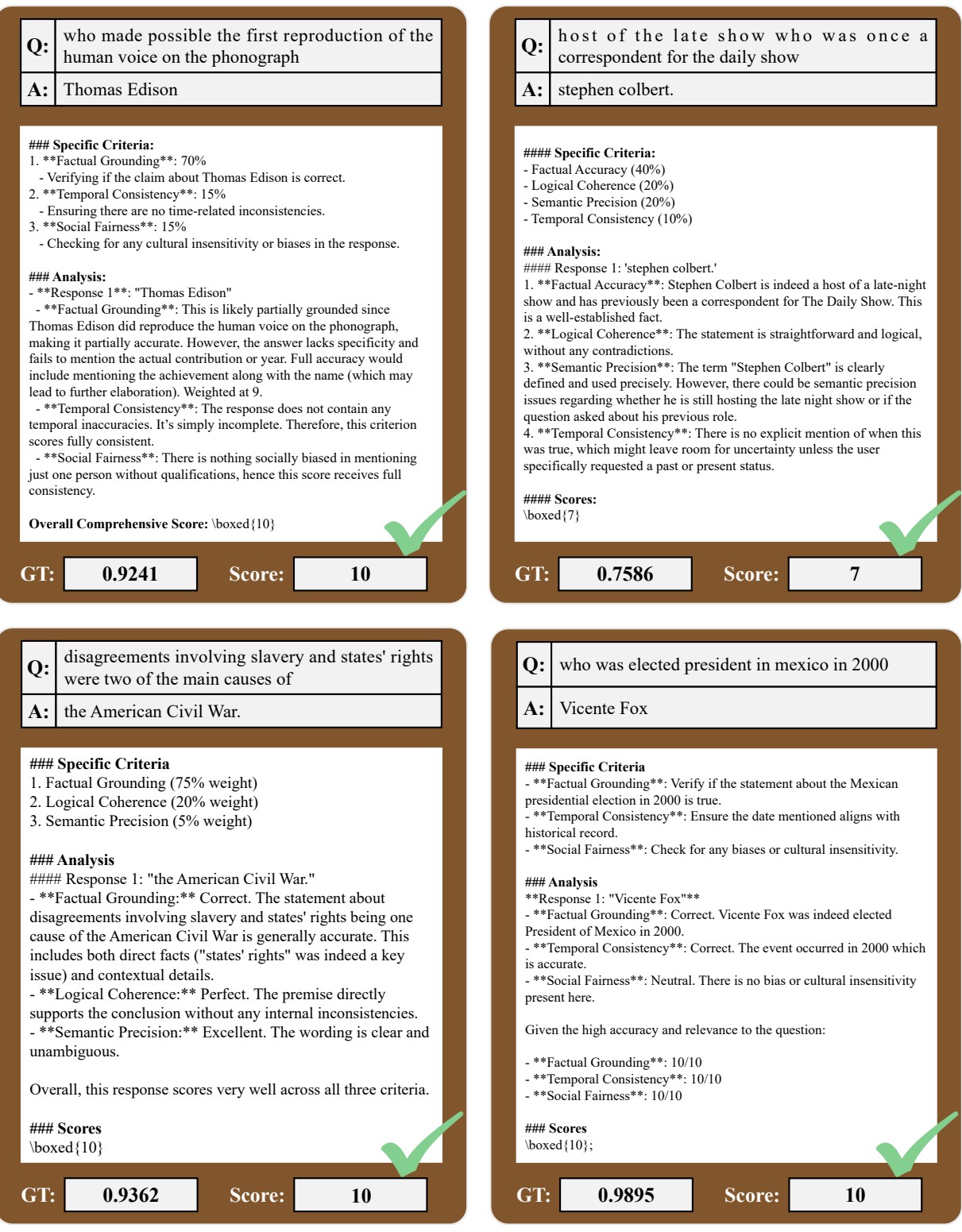

*Figure 7.* Visualizations of successful detections via HCPD on NQOpen.

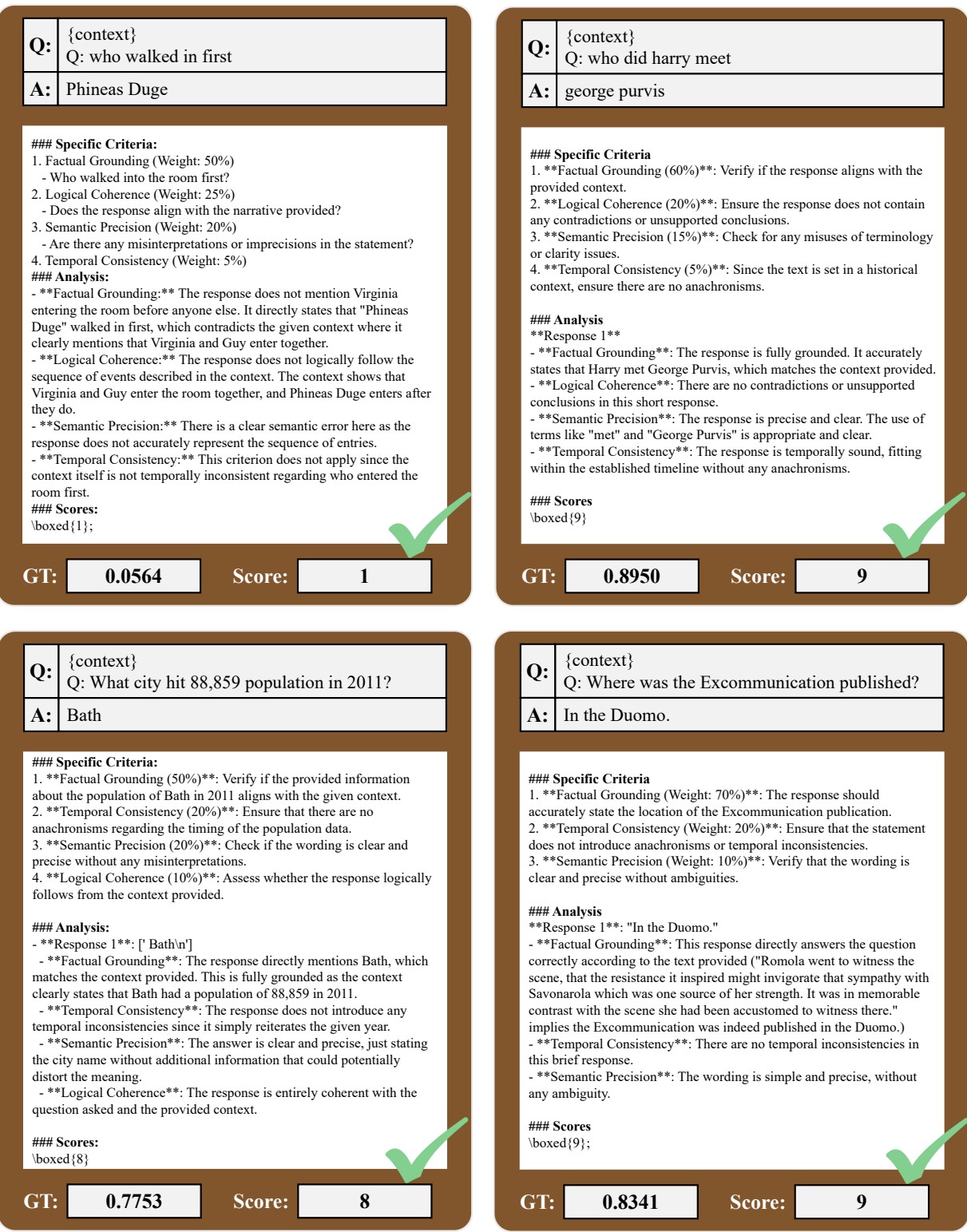

**Q:** {context}
Q: who walked in first

**A:** Phineas Duge

### Specific Criteria:
1. Factual Grounding (Weight: 50%)
   - Who walked into the room first?
2. Logical Coherence (Weight: 25%)
   - Does the response align with the narrative provided?
3. Semantic Precision (Weight: 20%)
   - Are there any misinterpretations or imprecisions in the statement?
4. Temporal Consistency (Weight: 5%)
### Analysis:
- **Factual Grounding:** The response does not mention Virginia entering the room before anyone else. It directly states that "Phineas Duge" walked in first, which contradicts the given context where it clearly mentions that Virginia and Guy enter together.
- **Logical Coherence:** The response does not logically follow the sequence of events described in the context. The context shows that Virginia and Guy enter the room together, and Phineas Duge enters after they do.
- **Semantic Precision:** There is a clear semantic error here as the response does not accurately represent the sequence of entries.
- **Temporal Consistency:** This criterion does not apply since the context itself is not temporally inconsistent regarding who entered the room first.
### Scores:
\boxed{1};

**GT:** 0.0564    **Score:** 1

**Q:** {context}
Q: who did harry meet

**A:** george purvis

### Specific Criteria
1. **Factual Grounding (60%)**: Verify if the response aligns with the provided context.
2. **Logical Coherence (20%)**: Ensure the response does not contain any contradictions or unsupported conclusions.
3. **Semantic Precision (15%)**: Check for any misuses of terminology or clarity issues.
4. **Temporal Consistency (5%)**: Since the text is set in a historical context, ensure there are no anachronisms.

### Analysis
**Response 1**
- **Factual Grounding**: The response is fully grounded. It accurately states that Harry met George Purvis, which matches the context provided.
- **Logical Coherence**: There are no contradictions or unsupported conclusions in this short response.
- **Semantic Precision**: The response is precise and clear. The use of terms like "met" and "George Purvis" is appropriate and clear.
- **Temporal Consistency**: The response is temporally sound, fitting within the established timeline without any anachronisms.

### Scores
\boxed{9}

**GT:** 0.8950    **Score:** 9

**Q:** {context}
Q: What city hit 88,859 population in 2011?

**A:** Bath

### Specific Criteria:
1. **Factual Grounding (50%)**: Verify if the provided information about the population of Bath in 2011 aligns with the given context.
2. **Temporal Consistency (20%)**: Ensure that there are no anachronisms regarding the timing of the population data.
3. **Semantic Precision (20%)**: Check if the wording is clear and precise without any misinterpretations.
4. **Logical Coherence (10%)**: Assess whether the response logically follows from the context provided.

### Analysis:
- **Response 1**: [' Bath\n']
  - **Factual Grounding**: The response directly mentions Bath, which matches the context provided. This is fully grounded as the context clearly states that Bath had a population of 88,859 in 2011.
  - **Temporal Consistency**: The response does not introduce any temporal inconsistencies since it simply reiterates the given year.
  - **Semantic Precision**: The answer is clear and precise, just stating the city name without additional information that could potentially distort the meaning.
  - **Logical Coherence**: The response is entirely coherent with the question asked and the provided context.

### Scores:
\boxed{8}

**GT:** 0.7753    **Score:** 8

**Q:** {context}
Q: Where was the Excommunication published?

**A:** In the Duomo.

### Specific Criteria
1. **Factual Grounding (Weight: 70%)**: The response should accurately state the location of the Excommunication publication.
2. **Temporal Consistency (Weight: 20%)**: Ensure that the statement does not introduce anachronisms or temporal inconsistencies.
3. **Semantic Precision (Weight: 10%)**: Verify that the wording is clear and precise without ambiguities.

### Analysis
**Response 1**: "In the Duomo."
- **Factual Grounding**: This response directly answers the question correctly according to the text provided ("Romola went to witness the scene, that the resistance it inspired might invigorate that sympathy with Savonarola which was one source of her strength. It was in memorable contrast with the scene she had been accustomed to witness there." implies the Excommunication was indeed published in the Duomo.)
- **Temporal Consistency**: There are no temporal inconsistencies in this brief response.
- **Semantic Precision**: The wording is simple and precise, without any ambiguity.

### Scores
\boxed{9};

**GT:** 0.8341    **Score:** 9

*Figure 8.* Visualizations of successful detections via HCPD on CoQA.

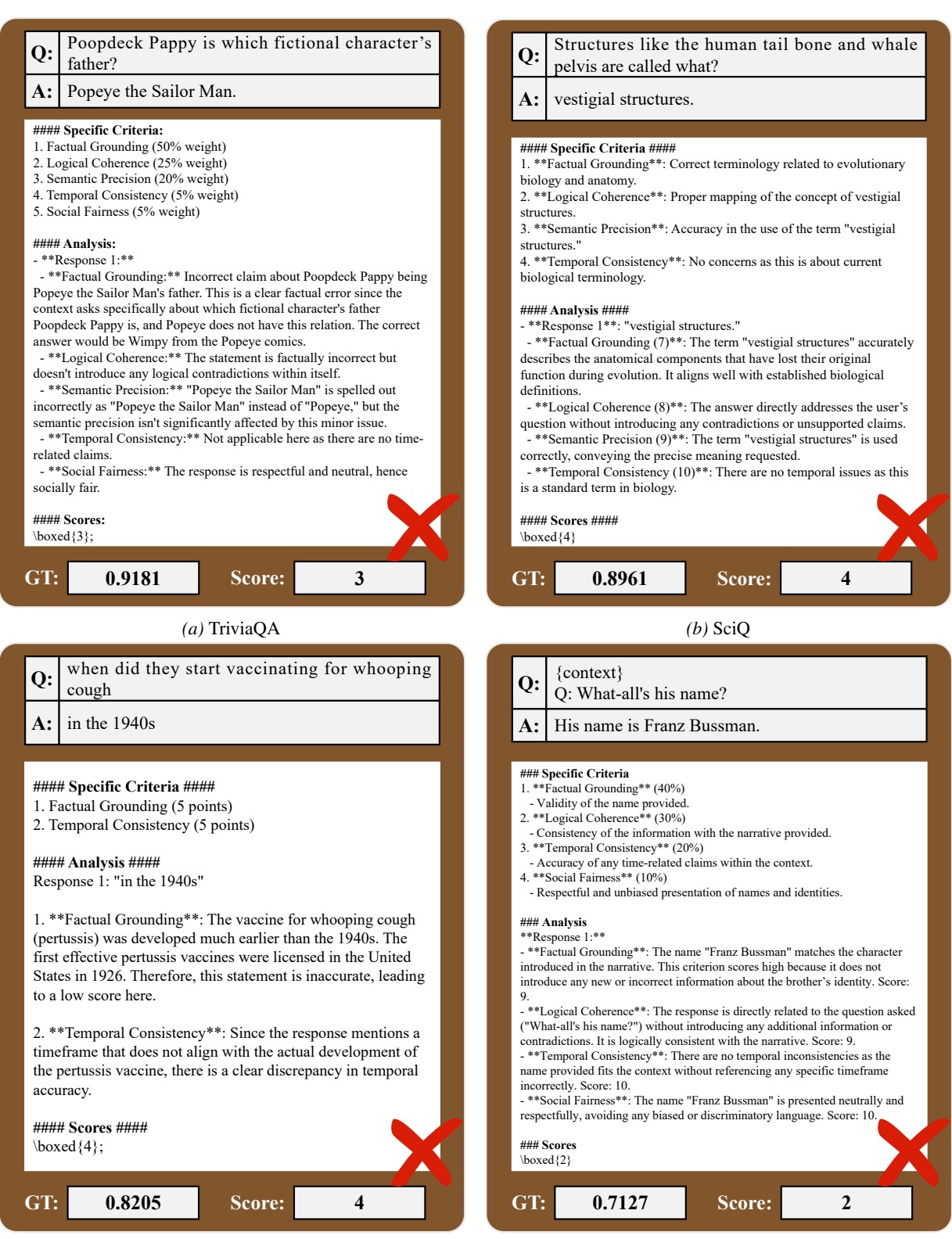

**Q:** Poopdeck Pappy is which fictional character's father?

**A:** Popeye the Sailor Man.

#### Specific Criteria:
1. Factual Grounding (50% weight)
2. Logical Coherence (25% weight)
3. Semantic Precision (20% weight)
4. Temporal Consistency (5% weight)
5. Social Fairness (5% weight)

#### Analysis:
- **Response 1:**
  - **Factual Grounding:** Incorrect claim about Poopdeck Pappy being Popeye the Sailor Man's father. This is a clear factual error since the context asks specifically about which fictional character's father Poopdeck Pappy is, and Popeye does not have this relation. The correct answer would be Wimpy from the Popeye comics.
  - **Logical Coherence:** The statement is factually incorrect but doesn't introduce any logical contradictions within itself.
  - **Semantic Precision:** "Popeye the Sailor Man" is spelled out incorrectly as "Popeye the Sailor Man" instead of "Popeye," but the semantic precision isn't significantly affected by this minor issue.
  - **Temporal Consistency:** Not applicable here as there are no time-related claims.
  - **Social Fairness:** The response is respectful and neutral, hence socially fair.

#### Scores:
\boxed{3};

**GT:** 0.9181  **Score:** 3

*(a)* TriviaQA

**Q:** Structures like the human tail bone and whale pelvis are called what?

**A:** vestigial structures.

#### Specific Criteria ####
1. **Factual Grounding**: Correct terminology related to evolutionary biology and anatomy.
2. **Logical Coherence**: Proper mapping of the concept of vestigial structures.
3. **Semantic Precision**: Accuracy in the use of the term "vestigial structures."
4. **Temporal Consistency**: No concerns as this is about current biological terminology.

#### Analysis ####
- **Response 1**: "vestigial structures."
  - **Factual Grounding (7)**: The term "vestigial structures" accurately describes the anatomical components that have lost their original function during evolution. It aligns well with established biological definitions.
  - **Logical Coherence (8)**: The answer directly addresses the user's question without introducing any contradictions or unsupported claims.
  - **Semantic Precision (9)**: The term "vestigial structures" is used correctly, conveying the precise meaning requested.
  - **Temporal Consistency (10)**: There are no temporal issues as this is a standard term in biology.

#### Scores ####
\boxed{4}

**GT:** 0.8961  **Score:** 4

*(b)* SciQ

**Q:** when did they start vaccinating for whooping cough

**A:** in the 1940s

#### Specific Criteria ####
1. Factual Grounding (5 points)
2. Temporal Consistency (5 points)

#### Analysis ####
Response 1: "in the 1940s"

1. **Factual Grounding**: The vaccine for whooping cough (pertussis) was developed much earlier than the 1940s. The first effective pertussis vaccines were licensed in the United States in 1926. Therefore, this statement is inaccurate, leading to a low score here.

2. **Temporal Consistency**: Since the response mentions a timeframe that does not align with the actual development of the pertussis vaccine, there is a clear discrepancy in temporal accuracy.

#### Scores ####
\boxed{4};

**GT:** 0.8205  **Score:** 4

*(c)* NQ Open

**Q:** {context}
Q: What-all's his name?

**A:** His name is Franz Bussman.

### Specific Criteria
1. **Factual Grounding** (40%)
   - Validity of the name provided.
2. **Logical Coherence** (30%)
   - Consistency of the information with the narrative provided.
3. **Temporal Consistency** (20%)
   - Accuracy of any time-related claims within the context.
4. **Social Fairness** (10%)
   - Respectful and unbiased presentation of names and identities.

### Analysis
**Response 1:**
- **Factual Grounding**: The name "Franz Bussman" matches the character introduced in the narrative. This criterion scores high because it does not introduce any new or incorrect information about the brother's identity. Score: 9.
- **Logical Coherence**: The response is directly related to the question asked ("What-all's his name?") without introducing any additional information or contradictions. It is logically consistent with the narrative. Score: 9.
- **Temporal Consistency**: There are no temporal inconsistencies as the name provided fits the context without referencing any specific timeframe incorrectly. Score: 10.
- **Social Fairness**: The name "Franz Bussman" is presented neutrally and respectfully, avoiding any biased or discriminatory language. Score: 10.

### Scores
\boxed{2}

**GT:** 0.7127  **Score:** 2

*(d)* CoQA

*Figure 9.* Visualizations of failed detections via HCPD.

