# OpenReview forum: "Zero-source LLM Hallucination Detection with Human-like Criteria Probing"
_ICML.cc/2026/Conference — ICML 2026 regular_

### Official Review · Reviewer_Xv23 · 2026-03-05

**Soundness:** 3
**Presentation:** 2
**Significance:** 3
**Originality:** 3
**Overall Recommendation:** 3
**Confidence:** 4

**Summary:**

This paper proposes HCPD (Human-like Criteria Probing for Hallucination Detection), a paradigm that detects hallucinations in LLM outputs under zero-source constraint, where no model internals or external knowledge bases access is involved. The core mechanism, named Human-like Criteria Probing (HCP), is introduced to drive an agent to adaptively generate interpretable criteria, assign context-aware weights, and produce evidence-grounded scores. Experiments across four datasets and several LLM families demonstrate substantial improvements over existing methods.

**Compliance With Llm Reviewing Policy:**

Affirmed.

**Final Justification:**

My main concerns have been addressed.

**Key Questions For Authors:**

- What is the principle to select five fixed criteria (Factual, Logical, Semantic, Temporal, Social) as the assessment standard?

- The final accuracy of the system relies heavily on the performance of the criteria weight assignment and scoring results. How to specifically confirm that the weight and the predicted scores of each criterion are correct?

**Limitations:**

The authors have not directly mentioned the limitations and potential negative societal impact of their work.

**Strengths And Weaknesses:**

### Strengths

- **S1**: Well-organized scoring mechanism. HCP involves multiple factors of criteria to assess the output of an LLM, which fits human judgment well. Multiple sampling is introduced, and the final score is ultimately aggregated, enhancing the robustness of the system as well.

- **S2**: Strong interpretability. The structured output format (criteria, weights, evidence, scores) provides transparency into why a particular verdict was reached. This is valuable and coherent for debugging and application， making the decision of the hallucination detection attributable.

- **S3**: Considerable experimental results. HCPD outperforms existing methods across multiple datasets, demonstrating its consistent advancements.

### Weaknesses

- **W1**: High computational cost. Sampling size is set as 5 in the experiment, which means that the probe needs to generate 5 full structured evaluations for each question-answer pair to detect the potential hallucination. The agent also requires training via GRPO in this paper. The entire cost seems prohibitive, which is not affordable for many real-world applications.

- **W2**: Weak supervision label quality. As mentioned in Section 4.4, the method relies on BLEURT scores as "weak supervision" labels during training, but BLEURT itself is a learned metric with many limitations. The paper assumes that BLEURT scores above 0.5 indicate "faithful" responses, but this threshold is arbitrary and may not align with human judgments.

- **W3**: Unimportant theoretical analysis. Section 4.6 involves several theoretical analyses illustrating the significance of some technical details, which are consensuses that have already been achieved. This section is encouraged to be put in the Appendix.

---

> ### Author Rebuttal · Authors · 2026-03-31
>
> > Q1. High computational cost. The overhead of GRPO training and multi-sampling inference (5 evaluations per instance) appears too expensive for practical deployment.
>
> A1. While we appreciate concerns regarding practical deployment, we respectfully clarify a misunderstanding about our computational overhead from 3 perspectives:
> * **Inference Time:** $K=5$ is not strictly required. As demonstrated in Tab. IV, HCPD outperforms the strongest baselines even at $K=1$, reducing inference latency to 0.2349s per sample.
> * **Inference VRAM:** Since most methods require LLM to extract internal states or log-probs, HCPD does not consume additional VRAM during inference. If evaluating massive models (70B mentioned by Reviewer pT7x), HCPD's fixed 7B footprint is more efficient.
> * **GRPO as a One-Time Offline Investment:** The RL alignment is an offline, one-time cost. Our cross-model and cross-dataset experiments confirm that once trained, the Agent operates as an off-the-shelf evaluator for texts from unknown sources.
>
> We provide a performance and consumption comparison as below, confirming HCPD matches the speed of light metrics and outpaces consistency-based methods.
>
> Table 1. Comparisons with baselines in terms of performance, inference time and GPU memory.
> ||AUROC|Inf. Time(s)|GPU Mem.(MB)|
> |-|-|-|-|
> |LN-Entropy|73.62|0.1383|17185|
> |CCS|78.20|1.1600|31498|
> |SelfCKGPT|74.58|6.5310|17540||
> |Perplexity|80.62|0.1349|17185|
> |SAPLMA|78.51|0.5341|16019|
> |Semantic Entropy|78.71|14.3026|17185|
> |Lexical Similarity|77.96|36.5646|17185|
> |EigenScore|51.35|37.7550|17453|
> |HaloScope|58.19|0.6012|16757|
> |TSV|79.78|0.0934|23592|
> |HCPD (Ours, $K=1$)|85.21|0.2349|18513|
> |HCPD (Ours, $K=5$)|86.25|1.1313|19051|
>
> > Q2. Weak supervision label quality. Relying on BLEURT with an arbitrary 0.5 threshold for weak supervision may not reliably align with human judgments of faithfulness.
>
> A2. We respectfully point out that our BLEURT and threshold setup is **a standardized setting for fair evaluation** [1][2], not an arbitrary assumption. We present empirical results with alternative metrics to confirm **HCPD's metric-agnostic nature**. We detail below:
>
> * **Rationale for BLEURT and 0.5 Threshold:** The choice of BLEURT and threshold strictly aligns with baselines (HaloScope, TSV) rather than being arbitrary. Since HCPD calibrate the Agent's 1-10 scoring rubric to the 0.5 anchor, isolated threshold shifts misalign the reward from the scoring instructions. Through joint calibration of rubric and threshold, HCPD operates robustly regardless of the threshold value.
> * **Generalization to Alternative Metrics:** BLEURT is an option, not a dependency. We conducte GRPO using alternative signals (ROUGE, DeepSeek-V3 as a Judge). Results **in Table 1 of Reviewer tpyD** confirm that as a metric provided, HCPD seamlessly aligns its behavior to reward signal and achieves strong performance.
>
> [1] HaloScope: Harnessing Unlabeled LLM Generations for Hallucination Detection. NeurIPS 2024.
>
> [2] Steer LLM Latents for Hallucination Detection. ICML 2025.
>
> > Q3. Unimportant theoretical analysis. Since the theoretical proofs in Section 4.6 represent established consensuses, they should be relocated to the Appendix.
>
> A3. We clarify the misunderstanding that our theoretical analyses uniquely **elucidate HCPD's design rationality** and **provide predictable error bounds**. Its value lies in mathematically formalizing the framework's internal mechanisms. Specifically, Theorem 1 links the GRPO objective to scoring behavior, while Proposition 1 and Corollary 1 prove how our fine-grained $[0,1]$ reward systematically bounds the deductive hallucination error across criteria.
> We will summarize a high-level theoretical intuition in the main text and move the redundant part to the Appendix.
>
> > Q4. What is the principle to select 5 fixed criteria as the assessment standard?
>
> A4. We clarify that our 5 criteria are grounded in established hallucination taxonomy [3], covering factual (Factual, Temporal) and faithful (Semantic, Logical, Social) hallucination. Our principle is to ensure exhaustive coverage of potential basic errors. Crucially, their application is not rigid. HCPD adaptively proposes context-specific criteria and weights based on the input.
>
> [3] A Survey on Hallucination in Large Language Models: Principles, Taxonomy, Challenges, and Open Questions. ACM TIS 2025.
>
> > Q5. Given their impact on final accuracy, how to specifically confirm the correctness of the generated criteria weights and predicted scores?
>
> A5. We clarify that intermediate correctness is strictly enforced by the sequential reasoning trajectory. Since the final score depends on the generated criteria, weights and rationales, any intermediate illusion inevitably distorts the result, incurring a severe penalty. Therefore, through extensive trial-and-error during RL, the Agent intrinsically learns a reliable evaluation mechanism. Our Case Studies (Figs. V-VIII in Appendix F) validate this logical behavior.

---

> > ### Author Rebuttal · Reviewer_Xv23 · 2026-04-04
> >
> > Thank you for your response, I have updated my score.

---

### Official Review · Reviewer_pT7x · 2026-03-10

**Soundness:** 2
**Presentation:** 3
**Significance:** 2
**Originality:** 2
**Overall Recommendation:** 4
**Confidence:** 3

**Summary:**

The paper proposes a new multi-criteria probing hallucination detection framework based on an LLM agent, which emulates the reasoning of a human expert via Human-like Criteria Probing. This method enables the LLM agent to decompose the judgment into interpretable criteria and aggregate them into a single nuanced score. The authors also propose an alignment training procedure using supervision derived from semantic consistency. Additionally, the authors provide an inference strategy with provable theoretical guarantees on the proposed alignment training policy. Finally, the authors demonstrate experimental evaluation across several tasks and LLMs.

**Compliance With Llm Reviewing Policy:**

Affirmed.

**Final Justification:**

The authors thoroughly addressed all concerns regarding the experimental evaluation, which substantially strengthens the paper. The main concern regarding the zero-source hallucination evaluation is partially addressed. However, it should be more clearly articulated in the paper.

**Key Questions For Authors:**

How are HaloScope and SAPLMA evaluated on a new model? Do they also use hidden states from the agent, or how are they evaluated? If not, how do they transfer between models given that they explicitly assume a fixed number of dimensions?

**Limitations:**

Yes

**Strengths And Weaknesses:**

**Strengths:**
1. Important and interesting approach towards more interpretable hallucination detection.
2. Consistent performance improvement in in-domain and out-of-domain evaluation across several QA tasks.

**Weaknesses:**
1. The main concern is regarding the claim of the zero-source hallucination detection approach. However, as stated, the method requires an LLM agent, e.g., Qwen-2.5-7B, which essentially means additional knowledge resources compared to the target model. Therefore, the claim of zero-source is somewhat overstated.
2. The paper evaluates experiments when the agent is approximately the same size as the target LLM, and therefore may have a similar knowledge base. It would be interesting to evaluate the method in cases where the agent is substantially smaller than the target LLM, e.g., 7B vs. 70B+.
3. Given the supervised nature of the method, it may be important to incorporate more state-of-the-art supervised baselines [1–3].
4. Generalization across tasks remains underexplored, e.g., trained on QA and evaluated on translation or summarization.


[1] Lookback Lens: Detecting and Mitigating Contextual Hallucinations in Large Language Models Using Only Attention Maps (Chuang et al., EMNLP 2024) \
[2] Unconditional Truthfulness: Learning Unconditional Uncertainty of Large Language Models (Vazhentsev et al., EMNLP 2025) \
[3] A Head to Predict and a Head to Question: Pre-trained Uncertainty Quantification Heads for Hallucination Detection in LLM Outputs (Shelmanov et al., EMNLP 2025)

---

> ### Author Rebuttal · Authors · 2026-03-31
>
> > Q1. The main concern is regarding the claim of the zero-source hallucination detection approach. However, as stated, the method requires an LLM agent, e.g., Qwen-2.5-7B, which essentially means additional knowledge resources compared to the target model. Therefore, the claim of zero-source is somewhat overstated.
>
> A1. We sincerely thank the reviewer for this insightful comment. We clarify that our "zero-source" claim strictly describes **the access constraints during inference** in real-world deployment. Particularly, it imposes two strict boundaries:
> * **No Access to Target Model Internals:** In real-world detection, the origin of the generated text is often unknown, or the target model is closed-source. Consequently, the detector cannot access its internal states, log-probabilities, or APIs.
> * **No Explicit External Retrieval:** For many specialized or emerging domains, searching for a suitable knowledge base is either impossible or computationally prohibitive.
>
> Under these severe constraints, an open-source model like Qwen2.5-7B functions as **a universally accessible**, offline, and fixed evaluator tool.
>
> > Q2. The paper evaluates experiments when the agent is approximately the same size as the target LLM, and therefore may have a similar knowledge base. It would be interesting to evaluate the method in cases where the agent is substantially smaller than the target LLM, e.g., 7B vs. 70B+.
>
> A2. We would like to point out that our manuscript already includes a cross-scale setting (Table III), where our Qwen2.5-7B Agent successfully evaluated larger target models (LLaMA2-13B, Qwen2.5-14B) and maintained strong generalizability.
> According to your suggestion, we conducted an experiment to evaluate hallucinations generated by LLaMA3.1-70B. In Table 1, HCPD continues to significantly outperform existing baselines, even when the target model is 10x larger:
>
> Table 1: Generalizability under hallucinations generated by LLaMA3.1-70B.
> |Method|Llama3.1-8b→Llama3.1-70b|Method|Qwen3-8b→Llama3.1-70b|
> |-|:-:|-|:-:|
> |SAPLMA|77.63±2.63|SAPLMA|78.14±1.64|
> |HaloScope|86.50±5.56|HaloScope|57.97±1.21|
> |TSV|81.29±10.23|TSV|58.85±5.70|
> |HCPD (Ours)|**86.87±0.98**|HCPD (Ours)|**89.73±2.63**|
>
> > Q3. Given the supervised nature of the method, it may be important to incorporate more state-of-the-art supervised baselines [r1–r3].
>
> > [r1] Lookback Lens: Detecting and Mitigating Contextual Hallucinations in Large Language Models Using Only Attention Maps (Chuang et al., EMNLP 2024)
>
> > [r2] Unconditional Truthfulness: Learning Unconditional Uncertainty of Large Language Models (Vazhentsev et al., EMNLP 2025)
>
> > [r3] A Head to Predict and a Head to Question: Pre-trained Uncertainty Quantification Heads for Hallucination Detection in LLM Outputs (Shelmanov et al., EMNLP 2025)
>
> A3. We sincerely thank the reviewer for pointing out these cutting-edge works. We have carefully reviewed them and will feature them in our revised "Related Work" section. We further evaluate one of the suggested baselines (LLM-TAD [r2]) as follows. In both tables, HCPD achieves superior performance compared to this SOTA baseline with white-box permission. Additional baselines will be added later.
>
> Table 2: Comparison of LLM-TAD and HCPD on Llama3.1-8b.
> |Llama3.1-8b|TriviaQA|SciQ|NQOpen|CoQA|Avg.|
> |-|-|-|-|-|-|
> |LLM-TAD|72.01±1.03|66.75±1.96|68.88±2.56|74.86±2.10|70.63|
> |HCPD (Ours)|**86.25±1.08**|**86.04±2.25**|**90.38±3.58**|**90.07±2.58** |**88.19**|
>
> Table 3: Comparison of LLM-TAD and HCPD on Qwen3-8b.
> |Qwen3-8b|TriviaQA|SciQ|NQOpen|CoQA|Avg.|
> |-|-|-|-|-|-|
> |LLM-TAD|71.27±1.90|55.98±4.37|76.82±3.77|66.94±0.58|67.75|
> |HCPD (Ours)|**93.69±0.89**|**92.63±2.90**|**87.35±6.22**|**84.80±1.01**|**89.62**|
>
> > Q4. Generalization across tasks remains underexplored, e.g., trained on QA and evaluated on translation or summarization.
>
> A4. We conduct an experiment on a Wikipedia Article Continuation task. In Table 4, HCPD maintains a performance advantage over existing baselines in the long-form setting. This confirms that our framework can robustly scale to long-form generation tasks.
>
> Table 4. Cross-dataset transfer performance on Wikipedia, using TriviaQA as the source dataset for training-based baselines.
> |Method|Wikipedia|
> |-|-|
> |LN-Entropy|73.02±1.76|
> |CCS|70.20±2.28|
> |SelfCheckGPT|72.37±2.03|
> |Perplexity|74.19±2.07|
> |HaloScope|71.74±1.30|
> |TSV|65.07±1.27|
> |HCPD (Ours)|**74.36±1.37**|
>
> > Q5. How are HaloScope and SAPLMA evaluated on a new model? Do they also use hidden states from the agent, or how are they evaluated? If not, how do they transfer between models given that they explicitly assume a fixed number of dimensions?
>
> A5. To evaluate their transferability and resolve the dimension mismatch, we use the source model (e.g., LLaMA3.1-8B or Qwen3-8B) as a surrogate feature extractor to encode text generated by an unseen target model. The obtained dimension-compatible representations are then passed to the baseline classifiers for evaluation.

---

> > ### Author Rebuttal · Reviewer_pT7x · 2026-04-03
> >
> > Thank you for the detailed rebuttal.
> >
> > Some of my concerns are addressed. However, the reported performance of the additional baseline (TAD) appears unexpectedly low compared to prior works and the other baselines presented in the paper. Specifically, this supervised method (TAD) performs on par with or worse than perplexity, which is quite surprising.
> >
> > Given the timing of the additional response in the rebuttal, it would be helpful if the authors could provide more details about the experimental setup, as well as any insights or explanations regarding the low performance, to better understand this discrepancy.

---

> > > ### Author Response · Authors · 2026-04-05
> > >
> > > We appreciate your insightful follow-up and are glad to provide the experimental details and our deeper insights to clarify this discrepancy.
> > >
> > > To ensure a rigorous comparison, we implemented TAD strictly using the **official codebase and default hyperparameters** [1], with the sole modification of replacing AlignScore with BLEURT to match our evaluation setting. Furthermore, we experimented with different aggregation strategies on TriviaQA and confirmed that the default settings are indeed optimal.
> > >
> > > Regarding the performance discrepancy, we hypothesize that it stems from a subtle misalignment between the training objectives of uncertainty regression and the hallucination detection. Methods designed to regress a sequence-level uncertainty score might not always optimally translate to tasks that require maximizing the separability between factual and hallucinated tokens. Under our evaluation settings, this difference in optimization focus could potentially limit their discriminative advantage over foundational metrics like perplexity.
> > >
> > > [1] https://github.com/mbzuai-nlp/llm-tad-uncertainty

---

### Official Review · Reviewer_tpyD · 2026-03-12

**Soundness:** 3
**Presentation:** 3
**Significance:** 3
**Originality:** 2
**Overall Recommendation:** 3
**Confidence:** 4

**Summary:**

This paper proposes Human-like Criteria Probing for Zero-source Hallucination Detection (HCPD), a method for detecting hallucinations in large language model outputs when only the query–answer pair is available, without access to model internals or external knowledge sources. The approach is to emulate how humans judge correctness by evaluating responses across multiple criteria rather than using a single score. The proposed mechanism generates a set of interpretable evaluation criteria (e.g., factual grounding, logical consistency, semantic accuracy), assigns adaptive weights to them depending on the context, scores the answer on each criterion, and aggregates them into a final truthfulness score. The detector is trained using a weakly supervised reward-based alignment scheme where scores are aligned with semantic consistency signals derived from reference answers, avoiding the need for manually labeled hallucination data. At inference time, the system performs multiple independent evaluations and averages the scores to reduce stochastic variance and improve reliability.

**Compliance With Llm Reviewing Policy:**

Affirmed.

**Key Questions For Authors:**

1. The training and evaluation both rely on semantic similarity metrics (e.g., BLEURT) as a proxy for hallucination severity. How sensitive is HCPD to the choice of similarity metric and threshold (e.g., 0.5)? Have you evaluated whether performance changes significantly when using alternative metrics or human-annotated hallucination labels?

2. The experiments are conducted primarily on QA datasets. Have you evaluated HCPD on other generation tasks such as summarization, long-form generation, or dialogue? If not, what challenges do you anticipate when applying the criteria probing mechanism to these settings?

3. Several existing works prompt LLMs to self-evaluate or critique outputs (e.g., self-consistency or self-check approaches). Could the authors clarify more precisely how the proposed human-like criteria probing differs from these methods, both conceptually and empirically?

4. The proposed method performs multiple evaluations per instance and aggregates the scores. How does inference latency scale with the sampling size in practical deployments, and are there strategies to reduce this overhead (e.g., fewer samples or adaptive sampling)?

**Limitations:**

yes

**Strengths And Weaknesses:**

**Strengths**

- The method is technically well motivated and combines a structured multi-criteria evaluation mechanism with weakly supervised RL alignment; experiments across multiple QA datasets and models show consistent improvements over several baselines, supporting the empirical claims.

- The paper is generally well organized, with a clear progression from motivation to method, theoretical analysis, and experiments, and includes illustrative examples that help explain the criteria probing mechanism.

- The work addresses the practical zero-source hallucination detection setting, where only query–answer pairs are available, which is relevant for real-world deployments involving black-box LLMs.

- The idea of modeling hallucination detection as human-like multi-criteria evaluation with adaptive weighting provides an intuitive and interpretable perspective that distinguishes the method from single-score or uncertainty-based detectors.

**Weaknesses**

- The training and evaluation rely on semantic similarity metrics (e.g., BLEURT) as proxy supervision for hallucination severity, which may not always correlate with factual correctness and could bias the results.

- Some sections, particularly the theoretical analysis and method description, are verbose and could be streamlined; distinctions from closely related self-evaluation approaches are not always clearly articulated.

- The evaluation focuses mainly on QA benchmarks, and it remains unclear how well the approach generalizes to other generation tasks such as summarization or dialogue.

- While the overall framework is interesting, many components (LLM self-evaluation, weak supervision, multi-sampling aggregation) are built from existing techniques, making the technical novelty somewhat incremental.

---

> ### Author Rebuttal · Authors · 2026-03-31
>
> > Q1. BLEURT may not reflect strict factual correctness. How sensitive is HCPD to this metric and the 0.5 threshold, and has it been validated against alternative metrics or human labels?
>
> A1. We appreciate this insightful feedback. We clarify that our BLEURT and threshold setup is **a standardized setting for fair evaluation**, not an arbitrary assumption. We present empirical results with alternative metrics to confirm **HCPD's metric-agnostic nature**. We detail below:
>
> * **Rationale for BLEURT and 0.5 Threshold:** The choice of BLEURT and 0.5 threshold strictly aligns with baselines (HaloScope, TSV) rather than being arbitrary. Since HCPD calibrates the Agent's 1-10 scoring rubric directly to this 0.5 anchor, isolated threshold shifts would misalign the reward from the scoring instructions. Through joint calibration of rubric and threshold, HCPD operates robustly regardless of the threshold value.
> * **Generalization to Alternative Metrics:** BLEURT is an option, not a dependency. We conducte GRPO using alternative signals (ROUGE, DeepSeek-V3 as a Judge). Results confirm that as an evaluation metric provided, HCPD seamlessly aligns its behavior to the reward signal and achieves strong performance.
>
> Table 1: Performance of HCPD and baselines under Alternative Metrics.
> ||Rouge|DeepSeek|
> |-|-|-|
> |LN-Entropy|77.70|52.69|
> |CCS|82.92|52.52|
> |SelfCKGPT|73.92|71.38|
> |Perplexity|85.49|53.82|
> |SAPLMA|87.22|79.99|
> |Lexical Similarity|82.40|53.63|
> |HaloScope|72.99|57.98|
> |TSV|78.46|73.31|
> |HCPD (Ours)|**89.17**|**80.42**|
>
> > Q2. Some sections are verbose and need streamlining. Furthermore, how does HCPD fundamentally differ from existing self-evaluation and self-consistency approaches, both conceptually and empirically?
>
> A2. We appreciate the feedback on the manuscript's flow. To streamline the paper, we will reorganize the theoretical analysis and transfer redundant parts to Appendix.
> And we clarify that while self-evaluation relies on heuristic zero-shot guessing or surface-level statistics, HCPD performs **adaptive, trajectory-optimized deduction** aligned via RL to diagnose structural flaws. We elaborate below:
> * **Conceptual Differences:** Self-evaluation methods rely on static prompts to elicit a zero-shot confidence score, and self-consistency methods measure statistical agreement across multiple generations. By contrast, HCPD utilizes an **adaptive template** for **explicit, multi-dimensional semantic deduction**. By **optimizing its entire reasoning trajectory** via RL, HCPD transforms from a heuristic guesser into a well-trained, context-aware evaluator.
> * **Empirical Differences:** The conceptual advantages translate into substantial empirical gains (Tab. I, Fig.III). HCPD outperforms self-consistency methods (e.g., SelfCheckGPT) and significantly increases through alignment.
>
> > Q3. How well does HCPD generalize from QA to summarization, dialogue, or long-form generation, and what challenges would criteria probing face there?
>
> A3. We conduct an experiment on a Wikipedia Article Continuation task. **In Table 4 of Reviewer pT7x**, HCPD maintains a performance advantage over existing baselines, confirming robust scalability to long-form generation.
>
> > Q4. While the overall framework is interesting, many components are built from existing techniques, making the technical novelty somewhat incremental.
>
> A4. We clarify that HCPD is not a mere tool assembly, but a conceptually novel framework tackling real-world deployment bottlenecks. For practical hallucination detection, a challenge is **the strictly bound "zero-source" constraint** that renders white-box methods inapplicable. Inspired by human expert cognition, we propose Human-like Criteria Probing Detector, which employs **an adaptive, multi-stage process** for highly accurate detection **relying solely on text**. Our contributions are threefold:
> * We introduce an **adaptive, multi-stage reasoning pipeline** instead of static prompts. By dynamically generating context-specific criteria and weights before scoring, HCPD elicits the LLM's internal knowledge to produce highly grounded judgments.
> * We map continuous reward into **a fine-grained 1-10 rubric** rather than binary classification. Fig. III confirms this alignment drives performance by offering richer supervision signals than standard methods.
> * HCPD produces **a structured, human-readable rationale** alongside the score, separating it from traditional approaches.
>
> > Q5. How does multi-sampling latency scale in practice, and what strategies (like fewer samples) can mitigate this overhead?
>
> A5. Tab. IV shows latency scales linearly with sample size $K$. We clarify that large $K$ is unnecessary as HCPD remains highly efficient at $K=1$. For latency-sensitive applications, "fewer samples" is a reasonable strategy.
> We provide a performance and latency comparison **in Table 1 of Reviewer Xv23**, confirming HCPD matches the speed of lightweight metrics and significantly outpaces consistency-based methods.

---

> > ### Author Rebuttal · Reviewer_tpyD · 2026-04-03
> >
> > The rebuttal partially addresses my concerns by showing that HCPD is not tied to BLEURT alone and by providing some evidence beyond QA. However, the method still relies on proxy supervision rather than direct factuality labels, robustness to thresholds remains insufficiently analyzed, and the distinction from prior self-evaluation approaches is clarified more by narrative than by controlled empirical decomposition.

---

> > > ### Author Response · Authors · 2026-04-04
> > >
> > > We sincerely thank the reviewer for acknowledging our additional experiments and for the continued constructive dialogue. We would like to address your remaining concerns below:
> > >
> > > > Q1. Method relies on proxy supervision rather than direct factuality labels.
> > >
> > > A1. We clarify that the severity of hallucination is hard to quantify using "direct factuality labels"  (i.e., 0 or 1), and that such annotation is burdensome even for expert annotators. Hence, established benchmarks [r1-r3] predominantly rely on consistency with true answers as a proxy. Moreover, HCPD operates via an intermediate reasoning trajectory (dynamically generating Criteria and Weights), where proxy supervision is **a methodological necessity** to cunstruct reward information and align the Agent's entire reasoning chain.
> > >
> > > [r1] Detecting Hallucinations in Large Language Models using Semantic Entropy. Nature 2024.
> > >
> > > [r2] HaloScope: Harnessing Unlabeled LLM Generations for Hallucination Detection. NeurIPS 2024.
> > >
> > > [r3] Steer LLM Latents for Hallucination Detection. ICML 2025.
> > >
> > > > Q2. Robustness to thresholds.
> > >
> > > A2. We wish to clarify a slight misconception: the 0.5 BLEURT threshold is **not a tunable hyperparameter**, but a **calibration standard defined by the benchmark** [r2, r3]. As detailed in Appendix C.2.2, HCPD’s 1-10 scoring rubric (e.g., 5: Noticeable factual errors，6: Minor inaccuracies, but core facts are correct) is strictly calibrated to this anchor. Shifting the threshold necessitates redefining the textual rubric itself, since they are intrinsically coupled. Crucially, our **DeepSeek-as-a-judge experiment** above entirely bypasses this issue, which **requires no threshold setting**. HCPD still achieves SOTA performance, empirically proving its success is independent of threshold engineering.
> > >
> > > > Q3. The distinction from prior self-evaluation approaches needs to be clarified by controlled empirical decomposition.
> > >
> > > A3. Following your suggestion, we conducted a controlled empirical decomposition against standard self-evaluation baseline [r4]. Results in Table 1 show that HCPD outperforms the baselines by Human-like Criteria Probing itself, and significantlt improved by our GRPO alignment.
> > >
> > > Table 1. Comparsion between self-evaluation approach and HCPD's components on TriviaQA for LLaMA3.1-8b.
> > > |Method|AUROC|
> > > |-|-|
> > > |Self-evaluation [r4]|56.07±1.92|
> > > |HCPD (Pre-RL)|66.54±0.59|
> > > |HCPD (Post-RL)|86.25±1.08|
> > >
> > > [r4] Language Models (Mostly) Know What They Know. Anthropic 2022.

---

### Official Review · Reviewer_TSMY · 2026-03-12

**Soundness:** 2
**Presentation:** 3
**Significance:** 2
**Originality:** 3
**Overall Recommendation:** 4
**Confidence:** 3

**Summary:**

This paper proposes Human-like Criteria Probing for Hallucination Detection (HCPD), where an LLM agent (i) generates evaluation criteria, (ii) assigns context-dependent weights, (iii) scores the answer per criterion, and (iv) aggregates the scores. The agent is aligned via GRPO with rewards derived from semantic similarity to reference answers, thereby avoiding human annotation. A multi-sampling strategy is used at inference. The effectiveness of HCPD is demonstrated by experiments on four QA datasets and six backbone models.

**Compliance With Llm Reviewing Policy:**

Affirmed.

**Final Justification:**

I'm satisfied with the author's response and thus increased my score

**Key Questions For Authors:**

Could the authors clarify the number of samples used for the sampling-based baselines, and training data and labels for the training-based baselines?

**Limitations:**

yes

**Strengths And Weaknesses:**

Strengths:

- The proposed HCPD framework achieves good empirical results without relying on any external knowledge.
- The per-criterion reasoning and weighting can provide interpretable signals to users.
- The analyses on expectation alignment and multi-sampling are helpful for understanding robustness.

Weaknesses:

- The paper motivates the zero-source setting by claiming that token probabilities and output distributions are unattainable for black-box and commercial LLMs. However, most LLM vendors, such as OpenAI and Antropic, offer access to log probabilities through API.

- The paper does not sufficiently explain its performance gain over baselines. Intuitively, black-box sampling-based approach should not outperform its counterparts that have access to log probabilities or model internal states (e.g., selfCheckGPT vs semantic entropy), especially considering that the agent is trained on a weak supervision signal derived from BLEURT-based semantic consistency scores.

- The proposed method requires RL training of the agent for each specific test dataset, which is quite resource-heavy. The cross-dataset transferability experiment results vary significantly across different models: for Llama-3.1, in-distribution training consistently produces the best results, whereas in the case of Qwen-3, training on other datasets and then transfer often produces even better results than in-domain training. As such, the reliability of these results is unclear.

- The evaluation is constrained to short-form QA, and it is unclear whether the method can generalize to long-form generation tasks, which are more common for practical applications.

- The evaluator agent is run 5–10 times per sample, which adds substantial overhead (\~1.1 seconds).

---

> ### Author Rebuttal · Authors · 2026-03-31
>
> > Q1. Major LLMs (OpenAI, Anthropic) offer API log probabilities, challenging the zero-source motivation.
>
> A1. We thank the reviewer for raising this point. While commercial APIs offer log-prob, a zero-source detector is necessary to **detect texts from unknown sources**, deal with **the limits of web interfaces**, and dispense with **the internal-state reliance of conventional baselines**. We detail the constraints below:
> * **Unknown Sources in the Wild:** In practice (e.g., audit web content), the source model of the generated text is often unknown, so querying for log-probs or internal states is impossible. The detectors rely solely on the text.
> * **Limitations of API and Web Interfaces:** Commercial APIs often truncate log-probs, hindering accurate uncertainty metrics. And most users interact via Web interfaces, providing outputs with no probability signals.
> * **SOTA Methods Require Internal States:** Most baselines require internal states. No commercial API exposes these proprietary representations.
>
> > Q2. The paper insufficiently explains how a black-box approach, supervised by BLEURT, can outperform baselines that access log-prob or internal states.
>
> A2. The performance gain of HCPD is attributed to 3 mechanisms: **multi-criteria decomposition** to isolate structural errors, **trajectory-level alignment** to fully utilize the supervision signals, and constraining the evaluation to **universal semantic space**. We detail the advantage below:
> * **Multi-dimensional Criteria Design Isolating Structural Errors:** HCPD mimics an expert by decomposing the evaluation into orthogonal dimensions, explicitly capturing nuanced errors that statistical metrics often miss.
> * **Trajectory-Level Optimization Utilizing Weak Supervision:** Moving beyond scalar regression, our RL alignment optimizes the entire reasoning trajectory. The agent is rewarded to generate context-adaptive criteria, weights, and rationales, enabling robust evaluation.
> * **Universal Semantic Space Ensuring Transferability:** HCPD operates in the natural language space, which is universally shared across LLMs, achieving superior transferability across diverse target models.
>
> > Q3. Per-dataset RL is resource-heavy. And the divergent cross-dataset transfer trends between LLaMA-3.1 and Qwen-3 make the results' reliability unclear.
>
> A3. HCPD requires only a **one‑time offline RL alignment**, avoiding per-dataset training. Cross-dataset experiments demonstrate good generalization, with results across 5 independent splits confirming stability. The divergent transfer behaviors between Llama3.1 and Qwen3 stem from reward quality: stronger target models construct a better RL reward landscape, allowing the agent to learn generalizable deductive logic.
> * **GRPO training is a one‑time offline investment, not dataset‑specific.** HCPD is a general‑purpose detector. As shown in Tab. III (cross‑model) and Fig. II (cross‑dataset), an agent trained on a single dataset achieves high AUROC on target datasets (81–87%) without finetuning.
> * **Cross‑dataset transfer results are statistically reliable, not anomalous.** All cross‑dataset and cross‑model results are averaged over 5 independent splits. The standard deviations are small (1–3% AUROC). The observation that training on CoQA sometimes outperforms in‑domain training is a stable pattern, not a random fluctuation.
> * **Explanation for the transferability discrepancy.** This difference likely arises from the reward landscape construction. Qwen3 generates plausible errors on complex tasks, providing dense, high-quality reward signals, enabling the Agent to learn generalizable deductive logic. In contrast, LLaMA3.1 struggles and yields rewards with less information. Both backbones maintain AUROC across transfers.
>
> > Q4. Unclear generalization to practical long-form generation tasks, as current evaluations are constrained to short-form QA.
>
> A4. We conduct an experiment on a Wikipedia Article Continuation task. **In Table 4 of Reviewer pT7x**, HCPD maintains a performance advantage over existing baselines, confirming robust scalability to long-form generation.
>
> > Q5. The evaluator agent is run 5–10 times per sample, which adds substantial overhead (~1.1 seconds).
>
> A5. We clarify that $K=5$ is not a strict requirement as HCPD remains highly efficient at $K=1$ (Tab. IV). We provide a performance and inference latency comparison as blow. **In Table 1 of Reviewer Xv23**, HCPD achieves comparable inference speeds to lightweight metrics and substantially faster than consistency-based methods.
>
> > Q6. Please clarify the sample sizes for sampling baselines, and the training data and labels for training baselines.
>
> A6. Sampling-based methods use a sample size of 10, following their original papers. Training-based methods adopt an identical 3:1 data split as HCPD for fair comparison. Labels are defined by BLEURT, adhering to the HaloScope and TSV setups. All official codebase URLs are listed in Appendix C.2 for reproducibility.

---

> > ### Author Rebuttal · Reviewer_TSMY · 2026-04-04
> >
> > Thank you for the detailed rebuttal and the new long-form generation experiments. While I appreciate the clarifications regarding training costs and inference efficiency, my core concerns regarding the practicality of the strict zero-source constraint and the stability of cross-dataset transferability remain.

---

> > > ### Author Response · Authors · 2026-04-05
> > >
> > > We sincerely thank you for continuing the discussion and for acknowledging our new experiments. We would like to address your remaining concerns below:
> > > > Q1. The practicality of the strict zero-source constraint.
> > >
> > > A1. Building on our previous discussion regarding the zero-source constraint, we further highlight the following prevalent real-world scenarios:
> > > * Social media platforms and news agencies audit massive volumes of user-uploaded text to filter out low-quality AI hallucinations or fake news. They rarely know whether a text is LLM-generated or which LLM it is generated by;
> > > * The vast majority of regular users interacting with LLMs through browser extensions or official web platforms (e.g., Claude [1], ChatGPT [2], Gemini [3]). These interfaces output pure text, stripping away any probability signals.
> > >
> > > Furthermore, even when commercial APIs are available, they present a significant performance bottleneck. They restrict users to lower-performing metrics like Perplexity, LN-Entropy, and Semantic Entropy (yielding AUROCs of 65.80% - 73.21%) and cannot support methods like SAPLMA or TSV (74.82% - 77.99%), which require internal states access.
> > >
> > > These real-world scenarios and empirical evaluations demonstrate that developing a **high-performing** hallucination detector that **relies solely on text content** (zero-source constraint) is both highly practical and fundamentally necessary.
> > >
> > > > Q2. The stability of cross-dataset transferability.
> > >
> > > A2. Regarding the cross-dataset transferability, we would like to clarify that fluctuations in absolute AUROC across different datasets are a **widely documented phenomenon**, rather than a limitation unique to our method.
> > >
> > > As observed in peer-reviewed works, OOD performance often varies unpredictably and can even exceed in-domain performance. For instance, HaloScope [4] trains on CoQA (76.42%) but achieves 77.36% on TruthfulQA and 92.98% on TyDiQA; TSV [5] trains on NQ Open (76.10%) and transfers to TruthfulQA at 76.90%; similarly, UHead [6] trains on English (46%) but tests higher on Russian (58%) and Chinese (54%).
> > >
> > > This phenomenon occurs because hallucination detection performance is not strictly dictated by ID vs. OOD distribution shifts. Instead, the absolute AUROC is a product of multiple confounding factors: the specific question types, the LLM's inherent competence (knowledge boundaries) regarding the dataset's topic, and the resulting supervision signal landscapes. Consequently, the absolute AUROC values naturally fluctuate across different benchmarks.
> > >
> > > To rigorously address this and ensure true algorithmic stability, we conducted multiple random data splits and independent train-test runs throughout our experiments. This statistical mitigation effectively eliminates the interference of data outliers.
> > >
> > > [1] https://claude.ai/new.
> > >
> > > [2] https://chatgpt.com.
> > >
> > > [3] https://gemini.google.com/app.
> > >
> > > [4] HaloScope: Harnessing Unlabeled LLM Generations for Hallucination Detection. NeurIPS 2024.
> > >
> > > [5] Steer LLM Latents for Hallucination Detection. ICML 2025.
> > >
> > > [6] A Head to Predict and a Head to Question: Pre-trained Uncertainty Quantification Heads for Hallucination Detection in LLM Outputs. EMNLP 2025.

---

### Decision · Program_Chairs · 2026-04-30

**Decision:**

Accept (regular)

**Comment:**

Summary:
This paper proposes Human-like Criteria Probing for Hallucination Detection (HCPD), a paradigm in which an LLM agent generates evaluation criteria, assigns context-dependent weights, scores the answers, and aggregates criterion-specific scores into a final truthfulness measure.  Experiments show that HCPD outperforms SOTA baselines when Llama-3.1-8b and Qwen-3-8b are used as the base LLMs.

Justifications:
Reviewer TSMY increased the score to 4 (weak-accept) after the rebuttal that resolved the reviewer's concerns. Reviewer tpyD did not acknowledge the follow-up rebuttal, so I assume the answers met the reviewer's expectations. Reviewer pT7x raised the score after several rounds of rebuttal discussion with the authors. Reviewer Xv23 also acknowledged that the authors' responses fully resolved the review concerns.